# Ventral midbrain stimulation induces perceptual learning and cortical plasticity in primates

John T. Arsenault [1,2,3] & Wim Vanduffel[1,2,3,4]

Practice improves perception and enhances neural representations of trained visual stimuli, a phenomenon known as visual perceptual learning (VPL). While attention to task-relevant stimuli plays an important role in such learning, Pavlovian stimulus-reinforcer associations are sufficient to drive VPL, even subconsciously. It has been proposed that reinforcement facilitates perceptual learning through the activation of neuromodulatory centers, but this has not been directly confirmed in primates. Here, we paired task-irrelevant visual stimuli with microstimulation of a dopaminergic center, the ventral tegmental area (VTA), in macaques. Pairing VTA microstimulation with a task-irrelevant visual stimulus increased fMRI activity and improved classification of fMRI activity patterns selectively for the microstimulation-paired stimulus. Moreover, pairing VTA microstimulation with a task-irrelevant visual stimulus improved the subject's capacity to discriminate that stimulus. This is the first causal demonstration of the role of neuromodulatory centers in VPL in primates.

[1] Laboratory for Neuro-and Psychophysiology, Department of Neurosciences, KU Leuven Medical School, 3000 Leuven, Belgium. [2] Massachusetts General Hospital, Martinos Ctr. for Biomedical Imaging, Charlestown, MA 02129, USA. [3] Leuven Brain Institute, KU Leuven, 3000 Leuven, Belgium. [4] Harvard Medical School, Boston, MA 02115, USA. Correspondence and requests for materials should be addressed to J.T.A. (email: john.arsenault@kuleuven.be) or to W.V. (email: wim@nmr.mgh.harvard.edu)

Visual perceptual learning (VPL) occurs when training at a visual task improves stimulus perception[1,2]. This type of learning has been shown to enhance the neural representations[3–5] specifically for the visual features that are relevant to the learnt task. In most documented examples of VPL, learning takes place when attention is oriented onto task-relevant visual features and reinforcement signals, either internal or external, are generated after task goals are achieved. The interaction of attentional and reinforcement signals has been hypothesized to drive VPL[6,7].

Neuromodulatory theories of VPL posit that behaviorally relevant events activate neuromodulatory centers[8,9] causing widespread neuromodulator release throughout cortex[10]. This diffuse signal then interacts with stimulus-driven activity in visual cortex[11] to generate plasticity ultimately leading to stimulus-specific VPL. Rodent studies have elegantly demonstrated stimulus-specific plasticity in visual cortex driven by activation of neuromodulatory centers[12,13], but these studies have not examined this with respect to perceptual learning nor selective attention. Attentional theories regarding VPL propose that selective top-down attention enhances relevant stimulus-driven activity while suppressing activity from irrelevant stimuli, thereby restricting plasticity to the attended stimuli[14,15]. Therefore, while attentional and neuromodulatory theories of VPL are mutually non-exclusive[7], a full understanding of the mechanisms of VPL requires the differentiation of these factors. Yet, separating neuromodulatory- from attention-driven effects is not straightforward because attention is commonly oriented toward relevant (e.g. reinforced) stimuli[16]. Task-irrelevant perceptual learning experiments offer an exception to this by demonstrating that Pavlovian stimulus-reward associations can generate VPL without attention oriented onto task-relevant stimuli[17,18]. These studies indicate that neuromodulatory signals generated by rewards may be sufficient to drive stimulus selective learning.

Therefore, to test neuromodulatory theories of task-irrelevant learning in primates, we designed experiments to test whether the association of a task-irrelevant visual stimulus and electrical microstimulation of the VTA (VTA-EM)[19], a neuromodulatory center in the midbrain, is capable of generating the physiological and behavioral effects that typify VPL. These experiments were designed to mimic task-relevant VPL paradigms whereby functional magnetic resonance imaging (fMRI) and behavior is measured before and after training. Importantly though, due to the task-irrelevant experimental design we employed, behavioral training was replaced with the Pavlovian association of an unattended visual stimulus and VTA-EM. Moreover, to ensure that the low-salience visual stimuli that were paired with VTA-EM were kept task-irrelevant, subjects performed a concurrent, difficult color discrimination task at a separate spatial location. Using this paradigm, we find that pairing task-irrelevant grating stimuli with VTA stimulation enhances the representation of these stimuli in visual cortex with the strongest modulations in posterior inferotemporal cortex (IT). We reveal in further experiments that associating task-irrelevant motion stimuli with VTA stimulation improves subject's ability to discriminate these stimuli and increased the fMRI response to these stimuli in posterior IT. Collectively, our findings demonstrate a causal role for the VTA in visual cortical plasticity and VPL.

## Results

**Confirmation of prediction error responses.** A microwire electrode array[20] was implanted into the VTA of both subjects using a peri-operative MRI guidance procedure[19]. To confirm the accuracy of the electrode positioning procedure, we recorded multi-unit activity (MUA) from subject M1 during a classical conditioning paradigm. In this paradigm 5 abstract visual stimuli (500 ms presentation) were each associated with a unique probability of juice reward (0–100% reward probability, juice delivered 400 ms after visual stimulus onset), while multi-unit activity (MUA) was recorded (Fig. 1a). MUA was found to be highly consistent across electrodes therefore results from a representative electrode are shown (see Methods). A stimulus detection response (50–100 ms, two-tailed signed-rank, $n = 23,538$, $z = 17.79$, $p = 8.51 \times 10^{-71}$) was first measured after stimulus presentation without a significant difference between reward probabilities (50–100 ms, Kruskal-Wallis, Chi-square = 5.14, df = 4, $p = 0.2731$). Afterward, MUA responses began to differentiate between reward probabilities with higher MUA responses for higher reward probability. Analysis of the time window 200–300 ms after stimulus response revealed a significant effect of reward probability (Kruskal-Wallis, Chi-square = 1211.4, df = 4, $p = 5.36 \times 10^{-261}$) and a significant difference between all reward probabilities (multiple comparison test, Tukey's HSD, $p < 0.01$, Fig. 1b). The effect of reward probability on the MUA responses (200–300 ms) was found during all sessions (Kruskal-Wallis, $p < 2.05 \times 10^{-10}$, Bonferroni corrected for multiple comparisons across 7 sessions). Moreover, lower reward probabilities (0 and 25%) exhibited decreased MUA relative to baseline activity, while MUA was increased for higher probabilities (50, 75, and 100%)

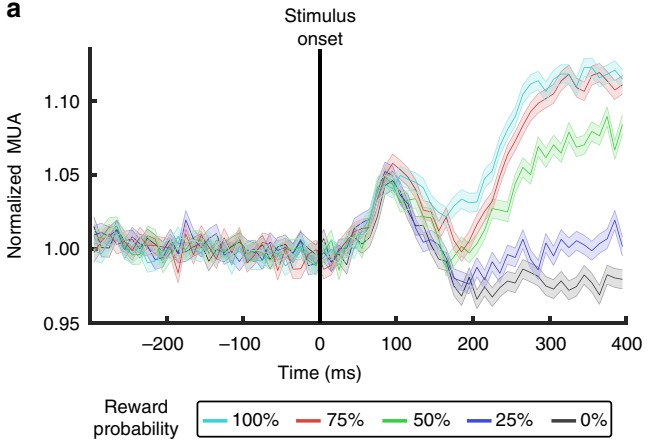

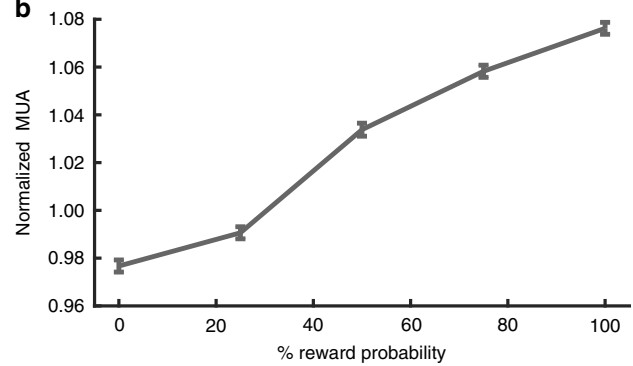

**Fig. 1** Electrophysiological confirmation of prediction error responses. **a** Peri-stimulus time histogram of normalized MUA recorded from a representative electrode from the chronic VTA array (see Methods). Visual stimuli were presented to subject M1 for 500 ms and associated with a unique reward probability (0, 25, 50, 75, 100%). Rewards were delivered after 400 ms based on stimulus-reward probability. **b** Mean normalized MUA 200–300 ms after stimulus onset. Error bars **a**, **b** denote sem across trials (0%, $n = 4823$; 25%, $n = 4706$; 50%, $n = 4669$; 75%, $n = 4672$; 100% $n = 4668$). Source data are provided as a Source Data file

during this time period (see Supplementary Table 1). These findings indicate that the array was positioned among VTA neurons encoding reward prediction errors.

**VTA-EM enhances representation of grating stimulus**. After confirming reward prediction error responses, we examined whether the association of a task-irrelevant visual stimulus with VTA-EM was sufficient to enhance the representation of that stimulus. Experiment 1 consisted of three sequential phases: a pre-association fMRI phase, a cue-VTA-EM association phase, and a post-association fMRI phase -with the fMRI phases being used to assess the changes in the grating stimulus representation resulting from VTA-EM. We first measured pre-association fMRI responses to noisy (60% noise pixels), low contrast (10%), oriented (45° or 135°) grating stimuli, that were presented in either the left or right visual field (LVF or RVF, Fig. 2a, Supplementary Fig. 1, and Methods). To orient attention away from the task-irrelevant stimuli (grating stimuli), monkeys performed a difficult color discrimination task with a small stimulus (0.22° × 0.22° square) presented in the upper visual field, at 7.5° deg eccentricity on the vertical meridian. Importantly, the color task was orthogonal to the grating stimuli with the color targets displayed on a given trial being independent from the grating stimulus shown. Colors were titrated to assure that the monkey's performance accuracy was kept below ceiling levels (Supplementary Table 2).

Individual trials of the pre-association fMRI experiment began with a randomized fixation period (Fig. 2b). Next, the color target and one of the four equiprobable grating stimuli were shown simultaneously for 500 ms. This was followed by a hand response to indicate color target identity (Supplementary Fig. 1). Monkeys used a left or right-hand response to indicate which of two colors were presented (more blue or red, respectively). Using this paradigm, the whole-brain fMRI responses to the different grating stimuli were mapped while attention was oriented onto the color task (see Methods).

After the pre-association fMRI data were acquired, the cue-VTA-EM association phase began (Fig. 2c and Supplementary Fig. 1). The design of these sessions was identical to the pre-association fMRI sessions with monkeys performing the relevant color discrimination task while task-irrelevant grating stimuli were presented concurrently. The one deviation from the pre-association fMRI phase was that one of the four oriented grating stimuli was associated with VTA-EM (100 Hz, 150–400 μA, see Supplementary Table 4). Because evidence suggests that delay conditioning creates relatively stronger associations between unconditioned and conditioned stimuli[21–23], the 200 ms train of VTA-EM began 300 ms after the onset of the grating stimulus (Fig. 2b). Therefore, the conditioned stimulus (i.e. grating stimulus) preceded and overlapped with the unconditioned stimulus (i.e. VTA-EM). After the cue-VTA-EM association phase was completed, post-association fMRI activity was measured using a protocol identical to the pre-association fMRI phase (Fig. 2c and Supplementary Fig. 1).

During all phases of experiment 1, color task difficulty was kept high with performance held at ~75–80% correct. Importantly, no differences in accuracy (Friedman's test, all phases $p > 0.2$, see Supplementary Table 2a) or reaction times (Friedman's test, all phases $p > 0.4$, see Supplementary Table 2b) on the color task were observed across conditions. This suggests that the attentional resources directed to the color task targets did not differ between the four task-irrelevant, low-salience grating stimuli. After examining color task performance, the effect of cue-VTA-EM association (post- vs. pre-association) on the orientation response (paired vs. control orientation) was assessed

using fMRI responses for stimulus presentations in the control VF (i.e. non-paired VF). No significant changes in the orientation response after pairing were found for the control VF (Fig. 2d). In contrast, cue-VTA-EM associations significantly increased the orientation response to the paired orientation when stimuli were presented in the paired VF (Fig. 2e). Enhanced activity was restricted to the hemisphere contralateral to the paired VF and overlapped in dorsal area V3 and PITv with the paired VF stimulus representation (light blue outline, as defined by an independent localizer, see Methods). To examine the specificity of the VTA-EM driven effects within different visual areas, we compared pairing-induced changes in the orientation response between the stimulus representations of the paired and the control VF. When stimuli were presented in the paired VF we found a trend for an interaction between VF representation and pairing in all visual regions. Although this interaction only reached significance in area PITv (Supplementary Fig. 2, three-way ANOVA, $p < 0.005$, Bonferroni corrected). This indicates that cue-VTA-EM association differentially modulated responses in the paired VF (increased paired orientation response) and control VF (decreased paired orientation response) representations with the strongest effects in area PITv. No significant interactions were found for stimuli presented in the control VF.

We next examined how cue-VTA-EM association altered the information in visual cortex needed to discriminate stimulus orientation using a searchlight multi-voxel pattern analysis (MVPA) throughout visual cortical regions (see Methods). Before cue-VTA-EM association, searchlight analyses for both the paired and control VFs did not reveal any regions with the capacity to classify the noisy, low contrast, irrelevant grating stimuli. After cue-VTA-EM association, we found significant orientation classification restricted to area PITv, contralateral to the paired VF, for stimuli that were presented in the paired VF (Fig. 3a, b, permutation test, $p < 0.005$, cluster size = 25 voxels, see Methods). This analysis did not reveal significant orientation classification for stimuli presented in the control VF after cue-VTA-EM association. These findings indicate that VTA-EM pairing selectively increased stimulus orientation information in the paired VF within area PITv.

**VTA-EM improves discrimination of motion stimuli**. In Experiment 2, we examined whether the association of a task-irrelevant cue and VTA-EM improved performance in a manner similar to perceptual learning. To avoid any cross-experiment contaminations due to stimulus exposure or implicit learning effects evoked during experiment 1, we examined perceptual learning in experiment 2 using a coarse motion discrimination task with completely different stimuli, stimulus locations and operant behavior relative to experiment 1. Each round of experiment 2 consisted of three phases: a pre-association motion discrimination phase, a cue-VTA-EM association phase, and a post-association motion discrimination phase -with motion discrimination performance used to assess microstimulation induced VPL (Fig. 4e).

During a given round of experiment 2, pre-association motion discrimination performance was tested utilizing random-dot motion stimuli (upward or downward motion) at five coherence levels (0–50%) presented in either the LVF or RVF (Fig. 4a, b, Supplementary Fig. 3, see Methods). Individual trials of the motion discrimination task began with a randomized fixation period (1500–2500 ms) followed by a 500 ms presentation of a motion stimulus. Animals then reported motion directions with a saccade onto one of two targets, either above or below the fixation point, to indicate upward or down motion, respectively. After the pre-association motion discrimination phase was used to

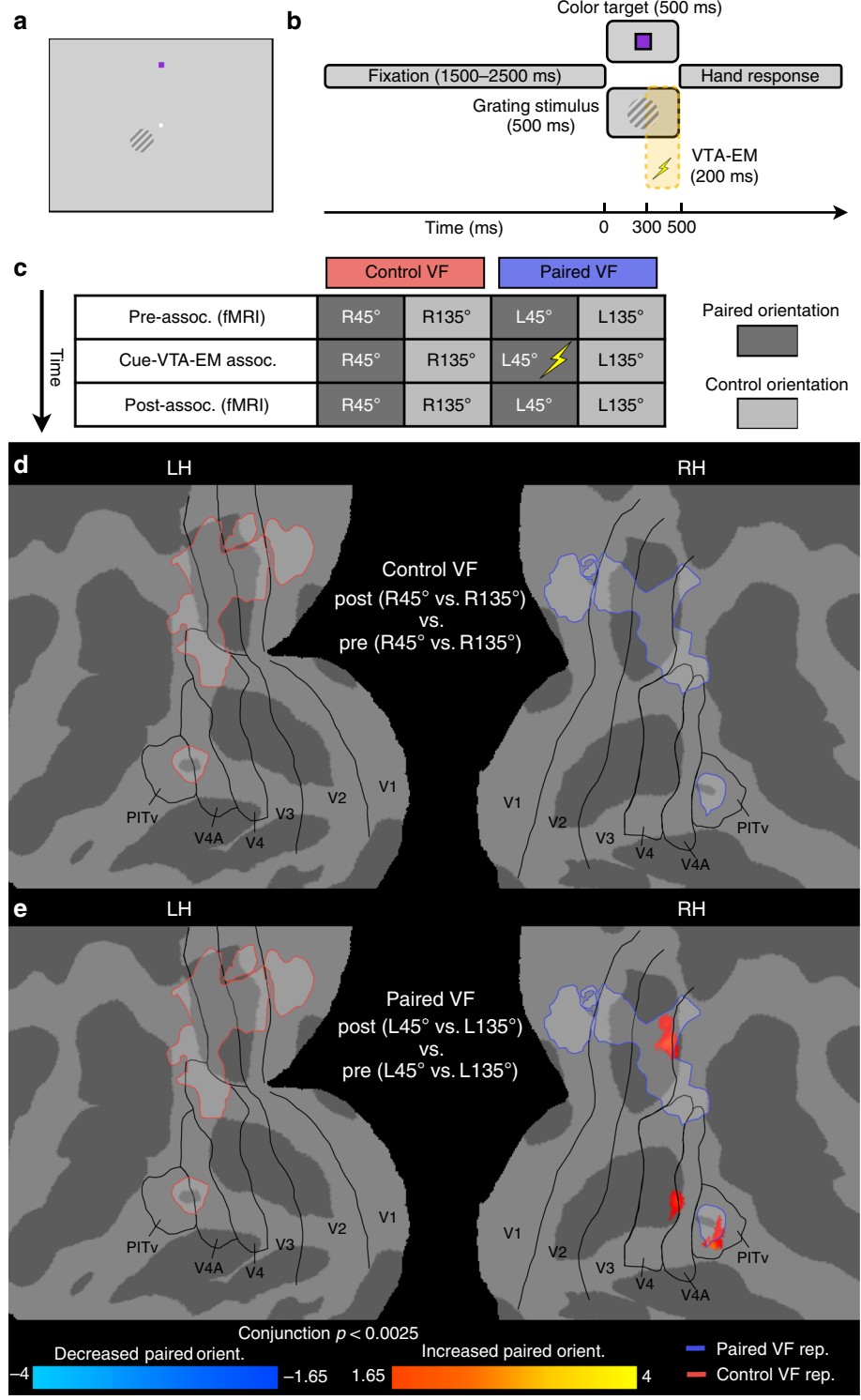

**Fig. 2** Experiment 1: VTA microstimulation selectively enhances stimulus representations. **a** Example positioning of color target (0.22° × 0.22° square, 7.5° above fixation point) and oriented grating in LVF (3° diameter, below (2.12°) and lateral (2.12°) to fixation point). **b** Timing of individual trials. After a fixation period (1500–2500 ms) a color target and an oriented grating (45° or 135°) in the LVF or RVF were shown simultaneously for 500 ms. The color of the target was reported with left- or righthand responses. During cue-VTA-EM association, the L45° stimulus was coupled with VTA-EM (100 Hz, see Methods). VTA-EM (200 ms duration) began 300 ms into the stimulus presentation (500 ms duration). **c** Chronological phases of experiment 1. **d**, **e** Flat maps depict significant differences (t-test, p < 0.0025, cluster size 40 mm², conjunction of M1 and M2) between post- and pre-association phases (M1–104 runs per phase; M2–78 runs per phase) in the orientation response (paired orientation vs. control orientation) for presentations to **d** control VF and **e** paired VF projected onto a flattened cortical representation. Black outlines depict visual areas defined by retinotopic mapping[72]. Light blue and red outlines depict the representation (independent localizer, general linear model (GLM), p < 0.001, M1 and M2 group data) of the grating stimuli presented in paired VF and control VF, respectively. LH = left hemisphere; RH = right hemisphere, L = left; R = right

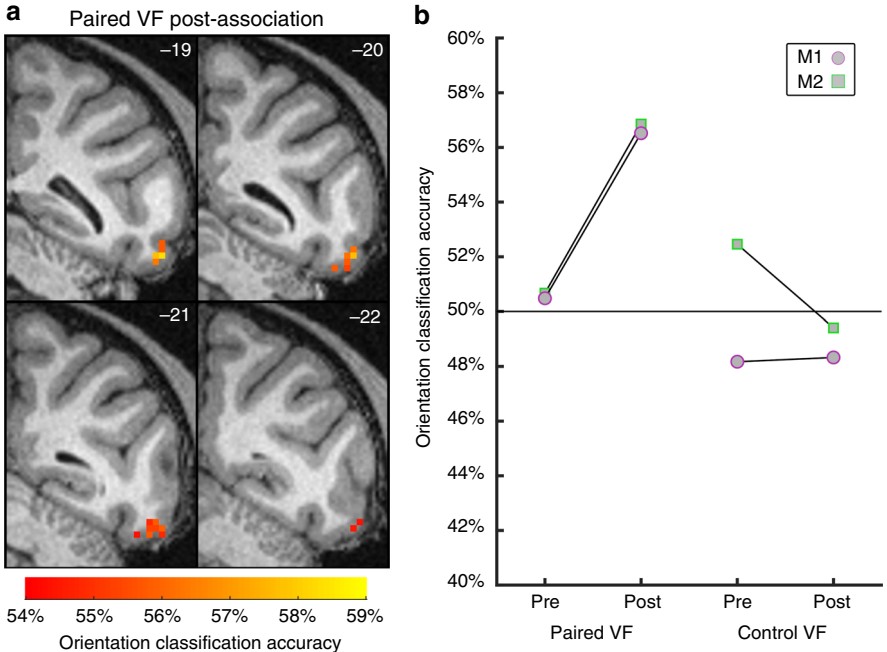

**Fig. 3** Experiment 1: VTA microstimulation selectively enhances stimulus information. A searchlight analysis (sphere of 20 voxels) using a naive Bayes classifier[73] was performed across all voxels in retinotopic visual areas (V1—PITv, see Methods). Searchlight analyses were performed separately for the pre- and post-association phases, the paired and control VFs and for each monkey (M1–104 runs per phase; M2–78 runs per phase). A cluster-corrected permutation test ($p < 0.05$, cluster size = 25 voxels, see Methods) on the conjunction of M1 and M2 data was used to determine significance. **a** Significant orientation classification accuracy for the paired VF in the post-association period is depicted on coronal slices (mm relative to inter-aural line denoted in top right). **b** Mean orientation classification accuracy within significant PITv cluster found from searchlight analysis of paired VF post-association data. Black line denotes mean orientation classification accuracy for the group. The purple circle and the green square with lines displays the mean orientation classification accuracy for M1 and M2, respectively

determine baseline performance, a series of cue-VTA-EM association sessions were performed (two monkeys, three rounds per monkey, see Supplementary Table 4).

During cue-VTA-EM association sessions, the relevant task was the color discrimination task employed in experiment 1 (positioned along vertical meridian) while the motion stimuli (positioned along horizontal meridian, Fig. 4c) were task-irrelevant for the monkey. Individual trials of the cue-VTA-EM association paradigm began with a fixation period (1500–2500 ms) followed by the concurrent presentation (500 ms) of a color target and an irrelevant 2% coherence motion stimulus (Fig. 4d, Supplementary Table 4). During a given round, only one of the motion directions (paired direction) in one of the VFs (paired VF) was associated with VTA-EM. A delay forward conditioning paradigm was also employed in experiment 2, therefore VTA-EM (200 ms duration) began 300 ms after the onset of the paired motion stimulus (500 ms duration). To avoid drawing attention away from the relevant color task during motion stimulus onset, random motion (0% coherence) was continuously presented at the potential motion stimulus positions in the LVF and RVF throughout the trial (Fig. 4d). Therefore, the task-irrelevant motion stimuli presented during the association phase are best described as an injection of exceedingly weak directional information into a continuous stream of random motion present in both hemifields.

To ensure that performance on the relevant color task during the cue-VTA-EM association phase did not differ between irrelevant motion stimuli, color task performance was assessed. No significant differences in accuracy (Supplementary Table 3a, Friedman's test, $n = 334$ runs, Chi-square = 4.05, df = 3, $p = 0.256$) or reaction time (Supplementary Table 3b, $n = 334$ runs, Chi-square = 2.29, df = 3, $p = 0.514$) were found, indicating that

attention on the relevant color task did not differ across motion stimuli. Next, we assessed the effect of VTA-EM on motion discrimination performance. Based on earlier task-irrelevant perceptual learning experiments[18], we hypothesized that if VTA-EM acted as a surrogate for reinforcement[19] then VTA-EM association should improve discrimination of weak, para-threshold stimulus strengths in the paired VF (see Methods). Consequently, we compared parathreshold motion discrimination sensitivity (d-prime, see Methods) during the pre- and post-association phases for stimuli presented to the paired or the control VF. By taking the control VF into account, we control for possible time-dependent reductions in performance resulting from the time between the pre- and post-association phases. A significant interaction of pairing and VF (linear mixed effect model, $F(1, 698) = 5.512$, $p = 0.019$, see Methods) was observed demonstrating that cue-VTA-EM association improved motion discrimination in the paired VF relative to the control VF (Fig. 4f). To assess this relative improvement in discriminative performance across the rounds of experiment 2, a sensitivity index (paired VF d′ - control VF d′) was calculated. This index increased after pairing in all rounds for both monkeys confirming the consistency of behavioral improvements for stimuli presented in the paired VF (Supplementary Fig. 4a). It is important to note that the monkeys did not perform the motion discrimination task for the duration of cue-VTA-EM association sessions plus weekend days that overlapped with these association sessions. A lack of recent task exposure tends to reduce task performance of monkeys, especially for difficult tasks. Therefore, it is not surprising that task performance was reduced in the control VF because of the long duration of time since the motion discrimination task was last performed. This also means that we are probably underestimating the true perceptual benefits

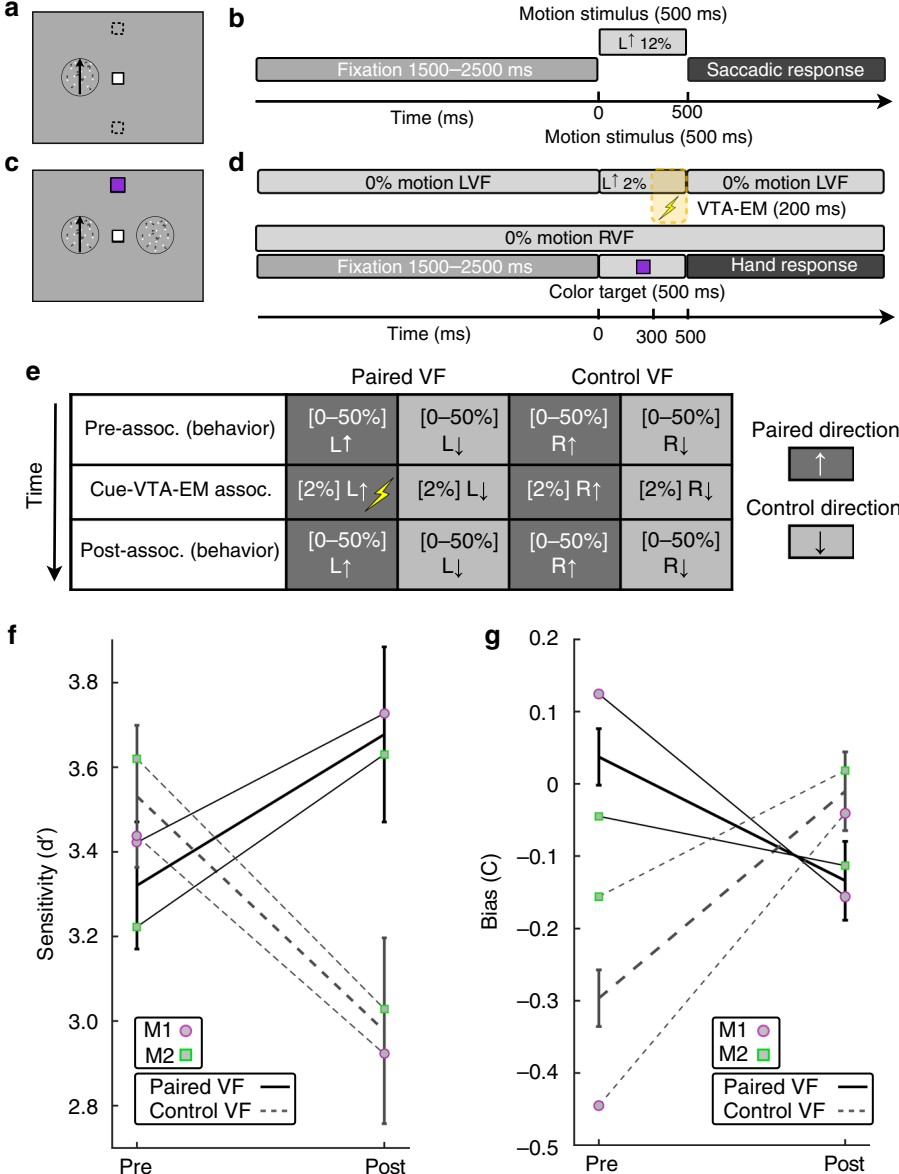

**Fig. 4** Experiment 2: Pairing improves stimulus detection and increases response bias. **a** Example positioning of LVF random-dot motion stimulus (4° diameter, 7.5° eccentricity) during the motion discrimination task. Dashed boxes represent virtual saccade response windows that were not visible during the experiment. **b** Timing of individual motion discrimination trials. After a fixation period (1500–2500 ms), a random dot motion stimulus (upward or downward) appeared for 500 ms. Motion direction was reported with a saccade. The coherence level (0, 2, 6, 12, 25, 50%), motion direction and VF in which the random-dot stimuli were displayed were equiprobable and randomized. **c** Positioning of stimuli during cue-VTA-EM association. Random-dot motion stimuli were positioned concurrently in the LVF and RVF. The color targets were identical to experiment 1. **d** Timing of individual trials of the cue-VTA-EM association phase. After a fixation period (1500–2500 ms), a color target was shown for 500 ms concurrently with a 2% motion stimulus in one VF and a 0% motion stimulus in the contralateral VF. A hand response reported color target identity. Before and after the color target, 0% motion stimuli were shown in both VFs. Only one condition (e.g. 2% left upward) was paired with VTA-EM (300 ms after motion onset, 200 ms, 100 Hz, Methods). **e** Outline of one round of experiment 2. The arrows indicate direction of motion. L = left VF, R = right VF. The range of coherence levels is indicated between brackets. Parathreshold **f** d-prime, and **g** c-criterion for 0–25% motion stimuli was compared between pre- (40 bins per round, 100 trials/bin) and post-association (20 bins per round, 100 trials per bin, see Methods) phases. Mean values for the paired VF (solid lines) and the control VF (dotted lines) during the pre-and post-association phases were calculated for the group (black lines) and individuals (M1—purple circle; M2—green square). Error bars denote sem for the group. Source data are provided as a Source Data file

induced by VTA-EM as we do not take into account the 'unlearning' that must have occurred in the paired VF.

We next hypothesized that if cue-VTA-EM associations also affected stimulus value then animals will likely become biased to report the VTA-EM-associated stimulus. To test this, we examined whether pairing altered the monkey's response bias (c-criterion, see Methods) with lower values of the c-criterion

denoting an increased bias to report the paired motion direction. We found a significant interaction between pairing and VF (linear mixed effects model, $F(1,47.40) = 16.688$, $p = 1.7 \times 10^{-4}$, see Methods) demonstrating that the bias to report the paired motion direction for stimuli presented in the paired VF increased after cue-VTA-EM association (Fig. 4g). A bias index (paired VF c – control VF c) was calculated to examine changes in bias across

rounds. We found that this bias index decreased during all rounds for both subjects indicating a consistent increase in bias to report the paired motion direction in the paired VF (Supplementary Fig. 4b). Therefore VTA-EM pairing consistently improved stimulus discriminability and increased bias to report the paired stimulus selectively within the paired VF.

**VTA-EM enhances representation of motion stimulus.** Finally, after exploring the behavioral effects of the Pavlovian association of VTA-EM with the low-SNR motion stimuli, we examined whether the representation of the paired motion stimulus was also affected by VTA-EM. To this end, experiment 3 comprised of three phases, with a pre-association fMRI phase, a cue-VTA-EM pairing phase, followed by a post-association fMRI phase. As in experiment 1, the fMRI phases were used to assess the changes in stimulus representations resulting from VTA-EM, but now for the random dot patterns (Supplementary Fig. 5, see Methods). The trial structure and general design of each phase was identical to the cue-VTA-EM association phase of experiment 2 with the exception that no VTA-EM was delivered during the pre- and post-association fMRI phases (see Fig. 4c, d, Supplementary Fig. 5). Therefore, the animals performed the orthogonal color task concurrently with the 2% motion stimuli in all phases of this experiment. Moreover, in all phases the short 2% coherent motion stimuli were embedded in a continuous stream of 0% coherent motion, while 0% coherent motion was also always present in the contralateral visual field. Both pre- and post-association fMRI responses to these weak motion stimuli was measured using this paradigm. To assess how cue-VTA-EM association affected the representations of the motion stimuli, we ran a second-level linear model on the direction response (paired direction vs. control direction) and examined the main effect of pairing (post- vs. pre-association) with VTA-EM (see Methods). For presentations of the 2% coherent motion stimulus in the control VF, no significant changes in direction response were found (Fig. 5a). In contrast, a significant increase in the paired direction of motion was found after cue-VTA-EM association within inferotemporal cortex, medial to PITv (Fig. 5b). This evidence corroborates the results of experiment 1 and indicates that, at least for the simple orientation and motion stimuli used, pairing with VTA-EM in a task-irrelevant manner specifically enhances stimulus representations in posterior regions of the temporal lobe.

## Discussion

We have shown that the association of VTA-EM and two entirely different task-irrelevant visual cues (i.e. gratings and random dot motion patterns) is sufficient to enhance the representation of these stimuli in primates. Moreover, pairing VTA-EM with an exceedingly weak, task-irrelevant motion stimulus selectively improved motion discrimination, demonstrating the causal role of primate VTA activity in task-irrelevant visual perceptual learning.

One of the defining features of VPL is its specificity with regard to the trained stimulus features, the position of the trained stimulus and even the trained eye[24–28] -although transfer has been observed in special circumstances[24,29]. The limited transfer of training benefits across features, locations and eyes, inspired early-stage models of VPL, which links plasticity to the earliest levels of the visual system[7]. However, despite the behavioral evidence pointing towards the importance of early visual cortical levels in VPL, training-induced changes at the single-cell level in primary visual cortex are much smaller than expected with an early-stage model[30–32] but see Yamahachi et al.[33]. In contrast, the significant plasticity observed at higher levels such as area V4 and

PIT cortex[34,35] argues for a mid-stage model of VPL[7], even when rather simple stimuli as gratings and random dot patterns are used. For example, an fMRI-guided electrophysiological study revealed that coarse orientation discrimination training increased the ability of PIT neurons to discriminate the trained stimuli[36], and considerably more so than in V4[37] and V1[32]. In the current study, we used fMRI to measure VTA-driven plasticity across visual areas. The enhanced response to the paired stimulus (based on general linear model analyses) and improved orientation classification (based on searchlight MVPA) was strongest in posterior IT areas. This indicates that posterior IT, and mid-stage visual areas more generally, may have a higher susceptibility to the VTA-driven component of VPL-related plasticity. This is also indirectly supported by a study examining reward-based learning of novel symbols, which showed plasticity in a similar region of posterior IT cortex[38]. It remains to be tested whether this generalizes to a wide range of stimuli and levels of discrimination (e.g. coarse versus fine). The stronger plasticity observed within these visual regions found with VTA stimulation and reinforcement learning may reflect the interaction of stimulus-driven activity and dopaminergic innervation within mid-stage regions. In addition, this hypothesis suggests that weak, unattended stimuli reduce the activity needed to generate plasticity in higher visual areas while sparser dopaminergic innervation in earlier visual regions renders these regions less conducive to plasticity. This is supported by studies of primate dopamine receptor distribution that have found more pronounced receptor densities in temporal compared to primary visual cortex[39]. In contrast, other neuromodulatory systems with stronger innervation to early visual regions[40] may also play a role in VPL-related plasticity observed in these earlier regions[33,41]. The latter hypothesis requires further testing.

Local plasticity in inferotemporal regions is one plausible mechanism through which performance could improve, yet other potential explanations must be examined. While we employed a difficult orthogonal task, and forward- and backward masked ultra-weak motion stimuli to avoid stimulus-directed attention during the VTA-EM association phase, preferential attention may still be allocated during the post-association behavioral testing of experiment 2. For instance, plasticity between typical attentional and visual brain regions could generate these effects[42]. Alternatively, acquired changes in stimulus value could also affect stimulus-specific allocation of attention accounting for the observed improvements[43]. Importantly though, with the sensitivity of fMRI, we did not find enhanced responses in fronto-parietal cortex, nor in subcortical regions such as the colliculus superior and typical valence-related regions, such as the nucleus accumbens and orbitofrontal cortex, as expected if attention was cued toward the paired cue. Therefore, local changes in inferotemporal cortex provide the most parsimonious explanation for the behavioral and functional changes we observed.

Pioneering rodent work has demonstrated that pairing stimulation of both dopaminergic and cholinergic neuromodulatory centers, with simple auditory stimuli increases stimulus representations in auditory cortex[44,45]. Moreover, these representational enhancements coincide with perceptual learning[46]. Further work has shown, that while increased stimulus representations correlate with improved stimulus perception, these enhancements dissipate over time, unlike the behavioral improvements[47]. This indicates that largescale representational enhancements, like those observed in the current study, may be a transient stage in the learning process that is later pruned into sparser representational enhancements. In reference to task irrelevant perceptual learning, the common use of anesthesia during rodent studies while pairing nucleus basalis stimulation[46,47] has provided perhaps the most compelling proof that Pavlovian association, in

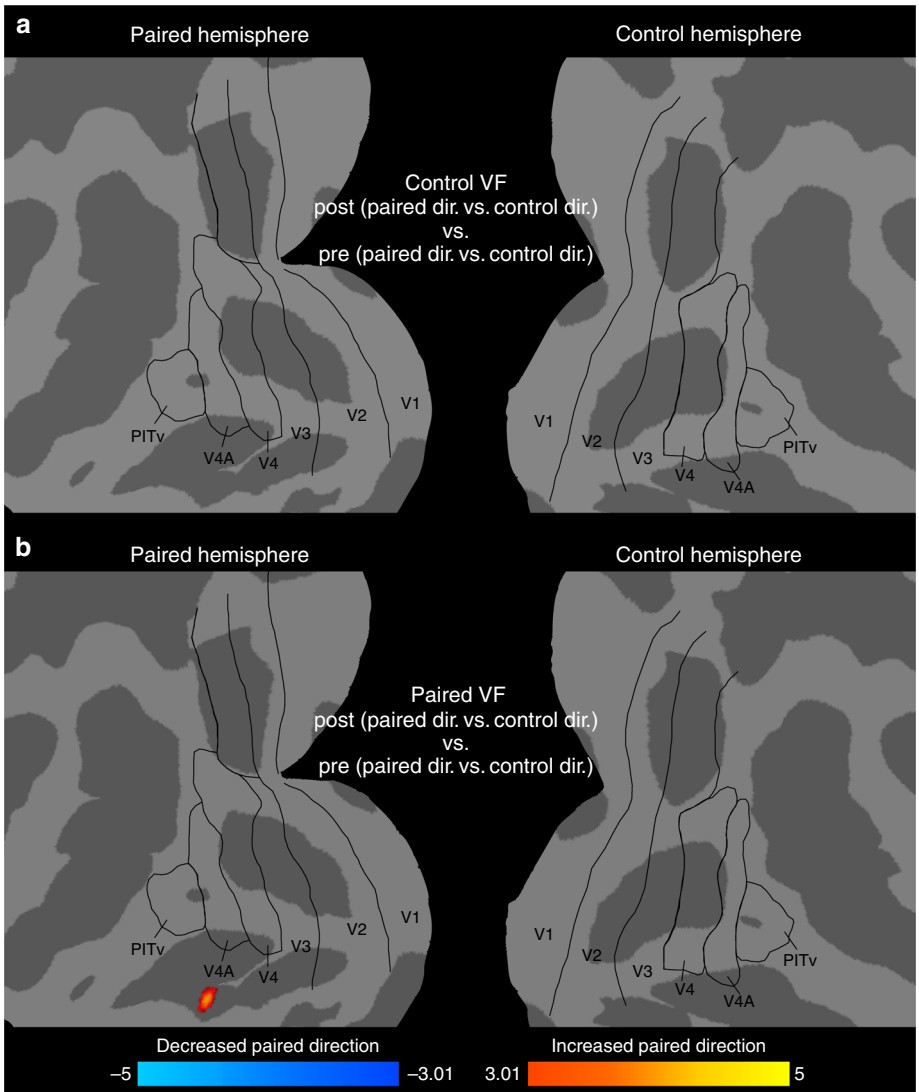

**Fig. 5** Experiment 3: Responses to weak motion stimuli selectively enhanced after VTA-EM. Flat maps depict main effect of pairing (second-level linear model, main effect of pairing – pre- vs. post-association, $p < 0.001$, cluster size 30 mm$^2$, M1–77 runs pre, 48 runs post; M2–47 runs pre, 43 runs post, see Methods) on the direction response (paired vs. control direction) for 2% coherent motion for presentations to **a** control VF and **b** paired VF projected onto a flattened cortical representation. Black outlines depict visual areas defined by retinotopic mapping[72]

the absence of attention, can generate perceptual learning. On the other hand, anesthesia may also interact with the normal functioning of the circuitry involved in different types of learning.

In the visual domain, long-term changes in rodent primary visual cortical responses have been found after reward pairing[48]. Further work has causally demonstrated that direct cholinergic input to area V1 drives the acquistion of these responses[12,13]. In addition to these longer-term changes, fast improvements in visual perception have been observed when cholinergic activity is acutely increased[49]. This indicates that the influence of neuromodulators on perceptual learning occurs across many timescales.

Across rodent studies examining the role of neuromodulatory regions, some of the most powerful designs utilize transgenic lines to target specific cell types. In contrast, considerable obstacles exist for cell-type specific modulations in primates (but see[50]). Therefore, despite earlier work demonstrating that primate VTA-EM elicits dopamine release[51,52], the mechanism through which primate VTA-EM affects behavior and physiology and whether that is dependent on dopamine alone remains to be determined.

Nonetheless, important species differences, like reduced visual acuity in rodents[53] and stronger cortical dopaminergic innervation observed in primates compared to rodents[39] suggests the mechanisms underlying VPL across species may vary substantially. In addition, during this study monkeys alternated between difficult visual tasks involving sustained fixation, peripheral attention and hand responses in a manner that was comparable to human studies of task-irrelevant VPL. Therefore, our work bridges the gap between rodent and human investigations of VPL.

In experiment 2, pairing VTA-EM with a task-irrelevant motion stimulus selectively improved motion discrimination, showing the causal role of VTA activity in primate task-irrelevant VPL. An increased bias towards reporting the paired direction was also found for the paired VF, demonstrating the capacity of phasic VTA activation to increase motivational value associated with a paired stimulus[19]. The combination of improved discrimination and increased value may reflect an important mechanism by which visual stimuli that consistently precede reinforcers, and subsequent VTA activation, are more easily

discriminated and more frequently approached. This may create a positive feedback loop whereby increasing stimulus detectability is coupled with increased stimulus preference. In turn these changes may increase the likelihood of further stimulus interactions, causing subsequent stimulus learning and increased stimulus preference. This process could lead to perceptual specialization, although such a process would depend on consistent stimulus-reinforcer relationships.

Higher-order cognitive components, like stimulus-directed attention, have been shown to play a key role in perceptual learning[14,15]. Despite this, it has been shown that VPL can also occur when subjects are not even aware of the stimulus carrying the feature being learned[54]. Since then, VPL of unattended stimuli has been demonstrated in a variety of paradigms utilizing different controls of stimulus attention and varying sources of reinforcement[17,55–57]. This line of evidence has led to the theory that task-irrelevant VPL can be driven solely by reinforcement signals, generated internally or externally, and that VPL may share the same neural mechanisms underlying classical conditioning[58]. Elegant optogenetic studies in rodents have shown that appetitive conditioning occurs when VTA dopamine neuron activation is temporally coupled with a sensory stimulus[59,60]. Based on such neural mechanisms, Roelfsema et al[6]. hypothesized that VPL is driven by the interaction of global neuromodulatory reinforcement signals (e.g. dopamine reinforcement signals) and visual stimulus activity.

In this study, we have demonstrated that association of a task-irrelevant stimulus with VTA activation is sufficient to drive stimulus-specific plasticity and increase stimulus discrimination sensitivity. This provides the first causal evidence that the activation of primate neuromodulatory centers can drive VPL, linking the neural mechanism of task-irrelevant perceptual learning with classical conditioning. It is also important to note that this Pavlovian component of perceptual learning that we examined using VTA-EM is only part of a typical perceptual learning process. Consequently, the small but consistent improvements we observed in d-prime are likely weaker than would be observed with a task-relevant design. Therefore, although there is a large degree of similarity between task-driven perceptual and procedural learning[61] and a likely overlap exists between the mechanisms generating these types of learning, our work adds to a growing body of evidence demonstrating a Pavlovian neuromodulatory component to perceptual learning.

In addition to phasic activity in the VTA, natural reinforcers activate cholinergic and serotonergic systems[62–64]. Therefore, VPL is likely gated by several neuromodulatory systems working in concert. Future work is needed to differentiate the roles of these neuromodulatory systems in driving the plasticity that underlies perceptual learning in primates.

## Methods

**Subjects**. Two rhesus monkeys (*Macaca mulatta*; M1, M2; 5–7 kg, 4–6-years-old, two males) were prepared for awake fMRI[65]. Animal care and experimental procedures were performed in accordance with the National Institute of Health's Guide for the Care and Use of Laboratory Animal, the European legislation (Directive 2010/63/EU) and were approved by the Animal Ethics Committee of KU Leuven. Animal housing and handling were according the recommendations of the Weatherall report, allowing extensive locomotor behavior, social interactions and foraging. All animals were group-housed (cage size at least 16–32 m³) with cage enrichment (toys, foraging devices) at the primate facility of the KU Leuven Medical School. They were daily fed with standard primate chow supplemented with bread, nuts, raisins, prunes and fruits. The monkeys were exposed to natural light conditions, with additionally 12 h of artificial light throughout the year, independent of the season. The animals received their daily water supply during the experiments until satiated.

**Electrical microstimulation**. Monkeys were fitted with MRI-compatible chambers (Crist Instruments) and a chronic electrode array was implanted using peri-operative MRI guidance[19]. The micro-brush electrode arrays[66] used to stimulate VTA consisted of 34 Pt/Ir microwires with polyimide insulation threaded through a 26 G microfil tube. The microwires of each electrode array consisted of a mix of 25 and 50 μm diameter wires. The microwires were attached to a 36-pin omnetics connector and were uniformly cut 5 mm past the microfil tube at the recording end. Unipolar electrical microstimulation was performed using the stimulating electrodes in the VTA and a low impedance ground wire implanted below the skull. The electrical microstimulation signal was produced with a stimulus isolator (DLS100, World Precision Instruments, current mode) connected to an eight-channel digital stimulator (DS8000, World Precision Instruments) and triggered by custom software that also controlled the visual and behavioral paradigms. Electrical microstimulation events were composed of stimulation trains lasting 200 ms and were composed of biphasic, square-wave pulses with a repetition rate of 100 Hz. Each pulse consisted of 0.2 ms positive and 0.2 ms negative voltage performed at 5 ms intervals. These stimulation parameters were used in experiments 1 and 2.

**Classical conditioning electrophysiology paradigm**. Trials were initiated when the monkey fixated on a central white fixation point. After a randomized fixation period (1000–2000 ms), one of the five possible abstract visual stimuli was randomly selected and presented for 500 ms. A juice reward delivery 400 ms into the presentation of visual stimulus was dependent on the probability of reward associated with that visual stimulus. Each visual stimulus was associated with a unique color, shape, position on the screen and reward probability (0, 25, 50, 75, 100%). Stimulus presentation and juice reward delivery timing were recorded with a photocell. Spiking activity was amplified and filtered to 500–5000 Hz. Neural activity and photocell pulses were digitized at 12,207 Hz (PZ5 Neurodigitzer, Tucker-Davis Technologies). Spikes were manually thresholded and sorted to remove non-spike waveforms and remaining spiking data was combined as MUA (Plexon Offline Sorter). Peri-stimulus time histograms from −300 to +400 ms were constructed for each stimulus (bins = 10 ms). MUA was normalized to the mean activity during the baseline period (−1000–0 ms). From all the electrodes that had sufficient SNR to record MUA (eight of 34), we present data from a representative electrode with higher SNR although responses to reward predicting stimuli were consistent across electrodes.

**Experiments 1 and 2 color discrimination task**. Monkeys were first trained on an orthogonal color discrimination task[36] with the purpose to disengage the animal's attention from the stimulus of interest (i.e. the gratings in experiment 1 and random dot patterns in experiment 2). Trials of this task began with a randomized fixation period followed by the presentation of one of two possible color targets for 500 ms. The color target (0.22° × 0.22° square) was positioned 7.5° above the central, white fixation point (0.14° × 0.14° square). After the 500 ms presentation of the color target, right hand responses were used to report color target A (more red) and left hand response were used to report color target B (more blue). Correct responses were followed by juice rewards. A left or right hand response was recorded when the monkeys hand interrupted an infrared light beam within the left and right hand box, respectively. The colors of the left and right hand targets were titrated such that performance was kept at ~75–80% correct responses.

**Experiment 1 oriented grating stimuli**. The circular, oriented grating stimuli (3° diameter) were centered at a position lateral (2.12°, either left or right) and below (2.12°) the central, white fixation point (0.14° × 0.14° square). Grating stimuli were comprised of sinusoidal gratings (2° per cycle, phase shifted in each trial) oriented at an angle of either 45° or 135° from the right horizontal axis. The orientation and the spatial location of the grating stimuli resulted in four stimulus types (L45°, L135°, R45°, R135°). The grating stimuli had a mean luminance of 75 cd per m² with 10% contrast between the lightest and darkest pixels and were presented onto a gray background (75 cd per m²). The stimuli were shown with 60% of the pixels in the grating stimulus being replaced with noise sampled from the sinusoidal luminance distribution.

**Experiment 1 color discrimination with oriented gratings**. Individual trials began with a variable fixation epoch (1500–2000 ms) in which the monkey had to maintain fixation on a central, white fixation point (0.14° x 0.14° square). Thereafter, a color target (described above in color discrimination task) and one of the four oriented grating stimuli (described above in oriented grating stimuli) were shown simultaneously for 500 ms. After the presentation of these stimuli, the animal reported the identity of the color target with a hand response. All four types of grating stimuli were shown with an equal probability. In addition, there was an equal probability for each grating stimulus to appear with either of the two color targets.

**Experiment 1 and 3 functional MRI acquisition**. Contrast-agent-enhanced functional images[65,67] were acquired in a 3.0 T horizontal bore full-body scanner (TIM Trio, Siemens Healthcare; Erlangen, Germany), using a gradient-echo T2* weighted echo-planar sequence (40 horizontal slices, in-plane 84 × 84 matrix, TR = 2 s, TE = 17 ms, 1.25 × 1.25 × 1.25 mm³ isotropic voxels). An external eight-channel phased array coil system (individual coils 3.5 cm diameter), with offline

SENSE reconstruction, an image acceleration factor of 3, and a saddle-shaped, radial transmit- only surface coil were employed[68].

**Experiment 1 fMRI analysis**. Images were first realigned using a non-rigid slice-by-slice registration algorithm[68] followed by SPM motion realignment. The resultant images were then non-rigidly co-registered. (www.nitrc.org/projects/jip)[69] to a template anatomical volume (M2) and resampled to 1 mm³. We then performed a voxel-based fixed-effect analysis with SPM5 for each individual run acquired during the pre- (M1–104 runs; M2–74 runs) and post-association (M1–104 runs; M2–74 runs) phases of experiment 1, and fit a general linear model (GLM)[65,67,70,71]. High- and low-pass filtering was employed prior to fitting the GLM. To account for head movement related artifacts, six motion-realignment parameters were used as covariates of no interest. To examine the effect of cue-VTA-EM association on the paired orientation response (paired orientation vs. control orientation), we used a second-level t-test. Specifically, we compared the paired orientation response per run between the post-association (M1–104 runs; M2–74 runs) and the pre-association (M1–104 runs; M2–74 runs) phases for presentations to the paired VF [Post-association (L45° vs. L135°) vs. Pre-association (L45° vs. L135°)]. The same analysis was done for presentations to the control VF [Post-association (R45° vs. R135°) vs. Pre-association (R45° vs. R135°)]. These analyses were performed separately for each animal and a conjunction analysis was performed across animals on the resultant maps (t-test, p < 0.0025, cluster size 40 mm²).

**Experiment 1 searchlight multi-voxel pattern analysis**. A searchlight analysis, to determine the accuracy of classifying between the paired and the control orientation, was performed throughout visual cortex (from V1 to inferotemporal cortex). Areas were defined from a co-registered probabilistic atlas of retinotopic visual regions[72]. This analysis was performed separately for stimulus presentations to the paired (left VF) and the control visual field (right VF) and separately for the pre- (M1–104 runs; M2–74 runs) and post-association (M1–104 runs; M2–74 runs) phases. This resulted in four separate searchlight analyses (paired VF pre-association, control VF pre-association, paired VF post-association, control VF post-association), which were performed for each monkey separately. Per run beta weights maps (see GLM analysis above) of the paired orientation (relative to fixation) and the control orientation (relative to fixation) for each of the analyses were used as inputs for a searchlight naive Bayes classifier as implemented in CoSMoMVPA[73] using a sphere of 20 voxels. For instance, during the pre-association paired VF searchlight analysis of M2, 74 per run beta maps of the contrast (L45° vs. fixation) and 74 per run beta maps of the contrast (L135° vs. fixation) were used as inputs. Leave-one-run-out cross-validation was employed and the average classification performance was used to determined the classification accuracy for the center of each sphere.

Statistical thresholds were determined using a permutation analysis. For each searchlight analysis, the labels were randomly reshuffled and an otherwise identical searchlight analysis was performed. This was repeated 10000 times. Based on the real classification accuracy and the accuracy found with the randomized label permutations, p values were defined for each voxel. A conjunction analysis was then performed across monkeys (permutation test, p < 0.005, cluster size of 25 voxels). To determine the cluster-corrected statistical significance of this searchlight analysis we again utilized a permutation analysis. P-values were calculated for each iteration of the randomized label permutation analysis based on the accuracy of all other iterations of the randomized label permutation analysis. A conjunction analysis was then performed across monkeys (p < 0.005) for each of the 10000 iterations and the probability of a cluster size greater than or equal to 25 voxels was determined. For each analysis the cluster-corrected threshold was p < 0.05 [paired VF pre-association (p = 0.0272), control VF pre-association (p = 0.0225), paired VF post-association (p = 0.0225), control VF post-association (p = 0.0228)].

**Experiment 1 color discrimination with grating stimuli**. Within experiment 1, the cue-VTA-EM association sessions were identical to the pre-association & post-association fMRI sessions with the exception that (1) the cue-VTA-EM association sessions occurred inside a mock scanner instead of the MRI scanner and (2) one of the four stimuli was temporally associated with VTA-EM. For both animals the L45° oriented grating was paired with VTA-EM. VTA-EM (200 ms duration, 150–400 µA, 100 Hz, two electrodes stimulated simultaneously) began 300 ms into the 500 ms grating stimulus presentation. The frequency and total number of cue-VTA-EM pairing events are detailed in Supplementary Table 4.

**Experiment 1 oriented grating stimulus localizer**. The oriented grating stimulus localizer was identical to the pre- and post-association phases of experiment 1 with the exception that the oriented grating stimuli were shown at a 50% contrast level. A voxel-based GLM analysis was performed on the individual runs of the localizer experiment for both animals (M1–65 runs; M2; 26 runs). To optimize our power to detect regions of visual cortex responsive to the oriented grating stimuli we combined data from both animals in a mixed effects analysis (i.e. a random effect analysis across runs) for the paired VF representation [(L45° + L135°)—fixation] and the control VF representation [(R45° + R135°)—fixation]. The resultant

statistical maps were thresholded (GLM, p < 0.001) and significant voxels that overlapped with a probabilistic atlas of retinotopic visual areas[72] were used in the ROI analysis of the pre- and post-association phases of experiment 1 displayed in Supplementary Fig. 2.

**Experiment 2 random dot motion stimuli**. The white noise random dot motion stimuli[74] were presented at a refresh rate of 60 Hz in a circular annulus (4° diameter) centered 7.5° degrees laterally (either to the left or right) of a central, white fixation point (0.14° × 0.14° square). Each dot had a diameter of 0.4° and the annulus contained 12.5 dots per degree². Between successive frames a percentage of dots determined by the motion coherence level of the respective stimulus (0, 2, 6, 12, 25 50%), were assigned as signal dots. These dots were positioned in the successive frame at a position dependent on the motion direction (upward or downward) and the speed of motion (5° per s). All other dots were classified as noise dots and randomly repositioned. Using these criteria new motion stimuli were generated for each stimulus presentation.

**Experiment 2 motion discrimination task**. Individual trials of the motion discrimination task began with a variable fixation epoch (1500–2000 ms) in which the monkey had to maintain fixation on a central, white fixation point (0.14° × 0.14° square). This was followed by 500 ms presentation of a random dot motion stimulus (described in random dot motion stimuli above) presented at 1 of 6 different motion coherence levels (0, 2, 6, 12, 25, 50%). After the presentation of these stimuli, the animal reported the direction of motion with a saccade. Upward saccades were used to report upward motion and downward saccades were used to report downward motion. On a given trial there was an equal probability of any of the four combinations of position and motion direction types (left upward, left downward, right upward, right downward). Each monkey performed three rounds of experiment 2, which included the pre- and post-association motion discrimination phases and the cue-VTA-EM association phase.

**Experiment 2 color discrimination with motion stimuli**. Individual trials began with a variable fixation epoch (1500–2000 ms) in which the monkey had to maintain fixation on a central, white fixation point (0.14° × 0.14° square). Thereafter, a color target (described above in color discrimination task) and one of the four possible 2% coherent motion stimuli (described above in random dot motion stimuli) were shown simultaneously for 500 ms. The identity of the color target was reported with a hand response. On a given trial there was an equal probability to present any of the four motion stimuli types (2% left upward, 2% left downward, 2% right upward, 2% right downward). Only 1 of the 4 motion stimuli was consistently paired with VTA-EM. VTA-EM (200 ms duration, 150–400 µA, 100 Hz, two electrodes stimulated simultaneously) began 300 ms into the 500 ms motion stimulus presentation. The frequency and total number of cue-VTA-EM pairing events are detailed in Supplementary Table 4. 0% coherent motion stimuli were always shown in both the LVF and RVF positions before and after motion stimulus onset (and also during the 2% motion stimulus presentation in the contralateral VF). Therefore, it is unlikely the animals consciously perceived the motion stimuli because of: (1) The use of two streams of random motion to mask the single weak motion stimulus greatly reduced the saliency of this already weak motion stimulus (2% coherence). (2) The animals were performing a difficult color discrimination task, drawing their attention to a spatial location at a large distance from the two potential locations of the weak, masked coherent motion stimulus.

**Experiment 2 d-prime and c-criterion calculation**. Hit rate and false-alarm rate were calculated separately for the paired VF and the control VF. Hits were defined as a correct response for presentations of the paired direction while false-alarms denoted an incorrect response for the control direction. Using these definitions, d-prime and c-criterion were calculated separately for bins of 100 trials. D-prime was calculated as the difference between z-scores of the hit rate and the false-alarm rates. C-criterion was calculated as the average of the z-scores of the hit rate and the false-alarm rates multiplied by −1. The c-criterion is an indicator of the bias to report the paired direction with more negative values indicating a stronger bias.

**Experiment 2 partitioning of behavioral data**. For each pre-test (n = 6 rounds, 2 monkeys × 3 rounds per monkey) the d-prime and c-criterion for each coherence level was calculated for 40 bins, each bin containing 100 trials. Therefore, for a given round, the 40 data points in the pre-association bins were taken from the 4000 trials occurring directly before the association round. For each post-test (n = 6 rounds, 2 monkeys × 3 rounds per monkey) the d-prime and c-criterion for each coherence level was calculated for 20 bins of 100 trials. Therefore, for a given round, the 20 data points in the post-association bins were taken from the 2000 trials directly after the association round—we took 2000 trials to estimate changes in performance directly after the association phase. From this partitioning, each round performed by each monkey (n = 6, 2 monkeys × 3 rounds/monkey) had 40 data points to describe the pre-association c-criterion or d-prime of a particular coherence level and 20 data points to describe the post-association c-criterion or d-prime of a particular coherence level.

**Experiment 2 partitioning of d-prime data**. From the general partitioning of data described above, we examined changes resulting from VTA-EM association for parathreshold levels of motion coherence. Parathreshold coherence levels of 12% and the 6% motion coherence level were utilized for M1 and M2, respectively. These motion coherences represented the lowest signal strength above 0% coherence levels where comparable pre-association motion discrimination sensitivity was found for both monkeys (mean pre-association d-prime value between 3 and 4).

**Experiment 2 analysis of d-prime data**. We constructed hierarchical mixed effects models[75] with monkey and the rounds each monkey performed as random effects factors and visual field (control or paired) and pairing (pre or post) as fixed effects factors. We compared a random slopes model to a random intercepts model using the Akaike information criterion (AIC) for model selection:

Interaction of visual field and pairing
random slopes model: $F(1,14.098) = 4.679$, $p = 0.0482$, AIC = 3257.83
random intercepts model: $F(1,698) = 5.512$, $p = 0.01917$, AIC = 3249.44
AIC values were lower for the random intercept model compared to the random slopes model, indicating the random intercept model was the most parsimonious.

**Experiment 2 partitioning of c-criterion data**. For the analysis of changes in c-criterion after VTA-EM, we examined all stimulus strengths except for the 50% motion coherence. In all, 50% motion coherence was excluded because animals performed close to perfectly at this coherence level leaving little room for changes in bias due to VTA-EM.

**Experiment 2 analysis of c-criterion data**. We constructed hierarchical mixed effects models[75] with monkey, rounds each monkey performed, and the snr levels displayed in each round as random effects factors and visual field (control or paired) and pairing (pre or post) as fixed effects factors. We compared a random slopes model to a random intercepts model using the AIC for model selection:

Fixed effects: Interaction of visual field (control or paired) and pairing (pre or post)
random slopes model: $F(1,47.402) = 16.688$, $p = 0.000169$, AIC = 11862.86
random intercepts model: $F(1,3462.8) = 22.954$, $p = 1.73*10^{-6}$, AIC = 11911.05
AIC values were lower for the random slopes model compared to the random intercept model, indicating the random slopes model is the most parsimonious model.

**Experiment 3 fMRI paradigm**. The experimental design of this fMRI paradigm was identical to the "Experiment 2: color discrimination task with concurrent motion stimuli (cue-VTA-EM association)" described above without VTA-EM and with the addition of a 0% coherence condition to provide a measure of baseline activity while no coherent motion was present. See Supplementary Fig. 5 for an overview of the fMRI paradigm.

**Experiment 3 color discrimination with motion stimuli**. This experimental design was identical to "Experiment 2 color discrimination with motion stimuli".

**Experiment 3 fMRI analysis**. Images were pre-processed and the fixed effects GLM fMRI analysis stream was identical to experiment 1 (see above). More specifically, a fixed-effect analysis was performed for each individual run acquired during the pre- (M1–77 runs; M2–48 runs) and post-association (M1–47 runs; M2–43 runs) phases of experiment 2. Individual data points in the analysis were comprised of the direction response (paired direction vs. control direction) from each run. Importantly during the cue-VTA-EM association phase M1 had downward LVF motion paired with VTA-EM while upward RVF was paired with VTA-EM for M2. Because visual information is preferentially processed in the contralateral hemisphere, before further analysis we flipped the hemispheres of M1. Therefore, data from the right hemisphere of M1 was positioned in the left hemisphere during analysis and consequently visual information from the paired VF (LVF) was now projected to the left hemisphere of the analysis space. This manipulation allowed us to combine data across monkeys while ensuring that the hemisphere contralateral to the paired VF was aligned across monkeys. We then examined the effect of cue-VTA-EM association on the direction response (paired direction vs. control direction). To accomplish this, we performed a second-level analysis using a linear regression model with factors pairing (pre or post-association) and monkey (M1 or M2) to control for individual differences in the direction response (paired direction vs. control direction). T-scores for the main effect of pairing are reported in Fig. 5.

**Statistical analysis**. For the analysis of MUA we analyzed responses 50–100 ms after stimulus onset. Each data point for these analyses was the mean MUA from 50–100 ms after stimulus onset during a single trial. A two-tailed signed-rank test was used to test for a stimulus response ($n = 23,538$) and a Kruskal-Wallis test was used to test for a difference in response between reward probabilities (0% $n = 4823$, 25% $n = 4706$, 50% $n = 4669$, 75% $n = 4672$, 100% $n = 4668$). We also analyzed responses during a later time window 200–300 ms after stimulus onset with

individual data points being comprised of mean MUA from this time window for a given trial. A Kruskal-Wallis test and multiple comparison test (Tukey's HSD) were used to compare MUA responses across reward probabilities (0% $n = 4823$, 25% $n = 4706$, 50% $n = 4669$, 75% $n = 4672$, 100% $n = 4668$). In addition, a Kruskal-Wallis test was also run for each session of the seven MUA sessions and results were Bonferroni corrected for multiple comparison across sessions.

For analysis of color discrimination performance in experiment 1, we used Friedman's test to compare reaction times and performance (percent correct) across conditions (pre-association, $n = 182$; cue-VTA-EM association, $n = 402$; post-association, $n = 182$). A Friedman's test was also used to compare reaction times and performance across conditions during experiment 2 ($n = 334$). Individual data points were comprised of mean reaction times or performance per run.

In the analysis of behavior during experiment 2, we examined both d-prime and c-criterion. For each pre-test ($n = 6$ rounds, 2 monkeys × 3 rounds per monkey) the d-prime and c-criterion for each SNR level was calculated for 40 bins, each bin containing 100 trials. For each post-test ($n = 6$, 3 rounds × 2 monkeys) the d-prime and c-criterion for each SNR level was calculated for 20 bins of 100 trials. Individual data points were comprised of the d-prime or c-criterion for one bin for stimuli presented to one of the visual fields. The lme4 R package[75] was used to generate linear mixed model.

**Reporting summary**. Further information on research design is available in the Nature Research Reporting Summary linked to this article.

## Data availability

The data reported in this paper are tabulated in the Supplementary Information and will be available upon reasonable request. In addition, the source data underlying Figs. 1a, b, 4f, g, Supplementary Fig. 4a, b and Supplementary Tables 1, 2a, b and 3a, b are provided with the paper as a Source Data file.

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

## Acknowledgements

We thank C. Fransen, C. Van Eupen, A. Coeman, P. Kayenbergh, I. Puttemans, C. Ulens, S. De Pril, A. Hermans, G. Meulemans, M. Depaep, W. Depuydt and S. Verstraeten and for technical and administrative support and S. Raiguel for his comments on the

manuscript. This work received funding from KU Leuven C14/17/109; Hercules II funds; Fonds Wetenschappelijk Onderzoek-Vlaanderen G0D5817N, G090714N, G088813N and Odysseus G0007.12; and the European Union's Horizon 2020 Framework Programme for Research and Innovation under Grant Agreement No 785907 (Human Brain Project SGA2). J.T.A. is a Postdoc fellow of the FWO.

## Author contributions

J.T.A. performed the experiments and analyzed the data with assistance from W.V.; all authors designed the experiments and wrote the paper.

## Additional information

**Competing interests:** The authors declare no competing interests.

