## [Transparent Peer Review File · Nature Communications]

Reviewers' comments:

Reviewer #1 (Remarks to the Author):

Here Arsenault and Vanduffel examine how electrical microstimulation of the ventral tegmental area (VTA-EM) could affect fMRI responses to a task-irrelevant visual input in macaque visual cortex. In Experiment 1, monkeys performed a color discrimination task while nominally-unattended oriented gratings were presented; one of four gratings was associated with VTA-EM. VTA-EM increased the fMRI response to that grating in contralateral dorsal V3 and PITv. In Experiment 2, upward vs downward coherent dot motion discrimination was assessed (again with the color discrimination task occurring during VTA-EM), and VTA-EM seemed to improve discrimination of the associated stimulus. These experiments are technically impressive and aim to provide important new data connecting neural changes to behavioral changes via activation of an important brain region (VTA) in non-human primates. However, at present I find the connections quite tenuous between the neural and behavioral changes; a lot more information is needed about experimental design, and the magnitude of the ultimate behavioral changes seems fairly weak, perhaps because the VTA-EM has not been optimized or is having off-target effects.

Could stimulation be cueing the animal to attend to the specific grating in some way, leading to enhanced discrimination abilities and consequent cortical plasticity by some other mechanism?

It's a bit inelegant to have different tasks for measuring cortical responses in Experiment 1 and reading out behavioral changes in Experiment 2. This considerably weakens the authors' attempt at providing the first information about perceptual learning and cortical plasticity with VTA-EM in primates, as the learning and plasticity relate to fundamentally different kinds of stimuli. (And thus there might be substantially different representational changes in Experiment 2 that provide the behavioral change; conversely there may be no consequence of the changes measured in Experiment 1.)

What makes dorsal V3 and PITv particularly sensitive to VTA-EM?

Does VTA-EM utilize dopamine, and/or co-release of some sort of factor to lead to these improvements? Any chance this involves fibers of passage along with or instead of direct VTA stimulation?

Studies of neuromodulation and plasticity generally indicate that the details of the stimulation association are critical in determining outcomes, perhaps especially so for VTA activation. The authors should clarify in the main text or figures how the stimulus was performed relative to visual stimulus presentation. What would happen if the temporal interval between VTA-EM and visual stimulus was altered? How did the authors choose their parameters for association and VTA-EM?

Were the multiunit recording studies in Figure 1 done for both monkeys, or just one? Either way this should be clarified; if both, the data shown separately for each.

It's not clear how the authors are using the word 'training'. I'd encourage them to strictly use this term to refer to behavioral training, and use some other term to refer to the VTA-EM procedure.

Effect size of VTA-EM is small, a change of d' of about 0.3 (or less than 10%).

Why does discrimination of stimuli in the control VF get worse by about the same magnitude as the 'trained' VF improves?

There are citations missing that provide critical background and context for these studies. The author are wrong in their central claims in the abstract and introduction that: 'no experimental evidence exists confirming the capacity of neuromodulatory regions to cause VPL and related adult

cortical plasticity in primates' and 'rodent studies have elegantly demonstrated stimulus-specific plasticity in sensory cortex driven by activation of neuromodulatory centers (Bao et al., 2001; Liu et al., 2015), but these studies have not examined this with respect to perceptual learning nor selective attention'. First note that those two references are not numbered and in the main reference list. Perhaps there's no one specific paper looking at visual perceptual learning and VTA stimulation with monitoring of cortical responses, but the authors' claim is still overblown in terms of the implications. I suggest the authors take a look and think about integrating other past work in primates (Liu et al. Current Biology 2017) and rodents (Froemke et al. Nature Neuroscience 2013; Livneh et al. Nature 2017; Ogg et al. Nature Communications 2018; Pinto et al. Nature Neuroscience 2013). I would say there's growing evidence connecting sensory cortical changes to specific enhancements in sensory processing and perceptual learning in different species.

Reviewer #2 (Remarks to the Author):

Arsenault and Vanduffel have designed two experiments that allow to clarify an important question, namely whether visual perceptual learning is driven by reinforcement and corresponding neuromodulatory centers or by attention-related mechanisms enhancing stimulus-driven activity. To this end they classically condition task-irrelevant stimuli with reinforcement provided by electrically activating the dopaminergic midbrain and examine whether this treatment results in changes in visual cortex fMRI activation and in improved perception tested in an operant task.

The first experiment consists of three phases. In the first phase they measure fMRI activation to task-irrelevant grating stimuli presented at different positions in the visual field while monkeys perform a visual task on different stimuli presented at other positions of the visual field. Subsequently one of the irrelevant grating stimuli was repeatedly paired with dopaminergic midbrain stimulation. This second phase was followed by a third phase repeating phase 1 and revealed a selective enhancement of the fMRI activation in the PITv to the irrelevant grating stimulus paired in phase 2. An additional searchlight analysis confirmed that only during phase 3 activity patterns in PITv contained sufficient information to classify the orientation of the grating stimuli.

The second experiment also consisted of three phases. In the first phase the monkeys performed a task different from that in experiment 1 and with different stimuli placed at different locations of the visual field. In phase 2, the monkeys performed the visual task of Experiment 1 while one the stimuli used in the task of phase 1 of Experiment 2 was repeatedly paired with dopaminergic midbrain stimulation. This treatment resulted, in phase 3, in enhanced performance of the task of phase 1 for the stimulus selected for classical conditioning in phase 2.

The experiments are very well designed and very well conducted, and the data are analysed with the state-of-the-art methods and using appropriate statistical tests. In general this research is highly relevant to a wider audience and the results are sound. At this stage, however, my enthusiasm about this research is somewhat dampened by the two issues detailed below.

The information given in the manuscript is highly compact. This makes it difficult for scientists not experts of visual perceptual learning to understand the manuscript. If the authors aim to publish their work in a non-specialist journal they should try to do a better job in describing their findings to a wider audience. In addition, they should relate their findings to the rich literature on auditory perceptual learning. Here, for example, it was shown that induction of cortical map plasticity by stimulation of the nucleus basalis improves learning (Reed A et al., Neuron. 2011 Apr 14; 70(1): 121-31; for review see also Irvine, D.R.F, Hearing Research 366, September 2018, Pages 3-16). They should specify how their work on visual perceptual learning advances over what is known on auditory perceptual learning and on procedural learning (Censor N et al. 2012. Nat. Rev. Neurosci. 13, 658e664).

It did not become clear to me how Experiment 2 was analysed. Unfortunately, the information given in the Legend of Figure 4 is very sparse ('40 bins/round, 100 trials/bin...') and the Methods are not fully helpful. Which trials and which sessions were used to determine the 'pre' and the 'post' motion discrimination sensitivity of the nonhuman primates? Did the authors combine the performance of Monkey 1 and Monkey 2? Did the authors average all pre-conditioning trials and compare them with the average of all post-conditioning trials? If this is true I wonder whether their analysis can contribute to the question whether their conditioning paradigm resulted in 'learning'.

Minor

There are some instances where the manuscript is poorly edited. For example, on page 3 references are not indicated by numbers. On page 4, the name 'Kruskal' is spelled in three different ways, the names 'Bonferroni' and 'Wallis' should also be corrected. On page 3, please explain in which way is the second task 'orthogonal' to the first task? Do you mean the two tasks are independent from each other?

The authors should provide more details how many leads of the multiwire array were used to test for prediction error responses, or whether they were combined into one multiunit record.

The authors should provide some reasoning why VTA-EM started 300 ms after visual stimulation, why 100 Hz was used and why this may be effective, in particular if one considers the conduction velocity of dopaminergic fibers. Was the stimulation unipolar or bipolar?

Reviewers' comments:

Reviewer #1 (Remarks to the Author):

Here Arsenault and Vanduffel examine how electrical microstimulation of the ventral tegmental area (VTA-EM) could affect fMRI responses to a task-irrelevant visual input in macaque visual cortex. In Experiment 1, monkeys performed a color discrimination task while nominally-unattended oriented gratings were presented; one of four gratings was associated with VTA-EM. VTA-EM increased the fMRI response to that grating in contralateral dorsal V3 and PITv. In Experiment 2, upward vs downward coherent dot motion discrimination was assessed (again with the color discrimination task occurring during VTA-EM), and VTA-EM seemed to improve discrimination of the associated stimulus. These experiments are technically impressive and aim to provide important new data connecting neural changes to behavioral changes via activation of an important brain region (VTA) in non-human primates. However, at present I find the connections quite tenuous between the neural and behavioral changes; a lot more information is needed about experimental design, and the magnitude of the ultimate behavioral changes seems fairly weak, perhaps because the VTA-EM has not been optimized or is having off-target effects.

1) Could stimulation be cueing the animal to attend to the specific grating in some way, leading to enhanced discrimination abilities and consequent cortical plasticity by some other mechanism?

The experiments described in our manuscript have distinct phases: phase-1: pre-testing (behavior or fMRI). phase-2: cue-VTA-EM association. phase-3: post-testing (behavior or fMRI). We feel that attention was not oriented onto the visual stimulus paired with VTA-EM during phase-2, the cue-VTA-EM association -which is the most critical phase of all experiments of this study. During the cue-VTA-EM association

phase, animals were performing a concurrent, difficult color discrimination task. Importantly, color task performance did not differ (see tables S1 and S2) between the VTA-EM paired and unpaired visual stimuli (both during the grating or motion experiments). This indicates that, even during the sessions that we actually stimulated the VTA, attention was focused on the difficult task and not diverted towards the visual stimulus that was paired with microstimulation.

The other important factor that helped ensure that paired stimuli were not attended was the low SNR levels of the grating and motion stimuli. The motion stimuli employed in experiment 2 were especially weak. The stimulus paired with microstimulation was a brief insertion of weak (2% coherence) motion information into a stream of random motion (0% coherence). In addition, random motion (0% coherence) was shown concurrently in the contralateral visual field. The animals would have to detect a weak signal from two streams of noise while performing a difficult orthogonal color task in order to be able to attend to the paired stimulus, which we suggest is exceedingly unlikely. To the best of our knowledge there is not a single study showing that monkeys are able to detect direction of motion in a 2% coherently moving stimulus that is positioned in the peripheral visual field. The additional orthogonal color task and the contralateral 0% coherent stimulus renders it virtually impossible that the monkeys pay attention to the short 2% coherent stimulus.

In contrast, during the post association behavioral testing of experiment 2, the monkeys were actively performing the motion discrimination task. Therefore, monkeys could preferentially attend to the previously paired grating. So, while it is unlikely that VTA stimulation drew attention of the animals during acquisition of the cue-VTA-EM pairing to a particular stimulus, this may occur during testing. However, the most important issue in relation to the learning component of VPL is the association phase of the experiment, not the testing phase.

Nonetheless, when examining how the Pavlovian association of VTA-EM and a task-irrelevant visual cue might improve stimulus discriminability there are several hypotheses with some of these explanations indicating increased attention toward the paired stimuli during testing. **Option A)** Dopaminergic input to visual regions may interact with visual activity to generate local plasticity within visual regions. This

mechanism would result from a Hebbian + dopamine like rule (i.e. neurons that fire together wire together if dopamine is co-released) (Bromberg-Martin et al 2010). This plasticity would then cause the selective enhancement of the visual cortical representation of the VTA-EM paired stimulus. In turn these representational changes lead to improved cue discrimination performance. This mechanism would rely on local processing not attentional cueing. **Option B**) Another possible mechanism through which VTA-EM could enhance the discrimination of cues paired with VTA-EM is through a feature-based attention-like mechanism. This could occur during phase-2 (cue-VTA-EM association) if VTA-EM is strengthening connections between stimulus representations in visual cortex and frontal (or other) attentional regions despite attention being focused on the orthogonal, color task at this time. Later during task performance, the improved frontally-driven signal would then increase visual cortical responses to the VTA-EM paired stimuli resulting in improved stimulus perception (Noudoost & Moore 2011). Although increased activity in visual cortical areas was found for paired stimuli, it is important to note that no increased activity in frontal (or other typical cortical and subcortical ‘attention’) regions were found for the paired stimulus -which would be expected in case option B was true. **Option C**) Finally, connections between limbic regions and visual cortical stimulus representations could also be enhanced. During post association testing changes in stimulus value could in-turn cue animals to increase attention towards paired visual stimuli.

While explanation **A** was focused on in our discussion of the original manuscript, we now have added discussion of explanation **B and C** with the caveat that, at least with the sensitivity of fMRI, we did not find enhanced responses in frontal and parietal cortex (and subcortical regions such as the colliculus superior) as we would expect if attention was cued toward the paired cue (see below). In summary, we still believe that scenario A is the most parsimonious explanation for our behavioral and functional results and that we obtained little evidence for supporting scenario **B and C**.

“Local plasticity in inferotemporal regions is one plausible mechanism through which performance could improve, yet other potential explanations must be examined. While we employed a difficult orthogonal task, and forward- and

backward masked ultra-weak motion stimuli to avoid stimulus-directed attention during the VTA-EM association phase, preferential attention may still be allocated during the post-association behavioral testing of experiment 2. For instance, plasticity between typical attentional and visual brain regions could generate these effects (Noudoost & Moore 2011). Alternatively, acquired changes in stimulus value could also affect stimulus-specific allocation of attention accounting for the observed improvements (Bourgeois et al 2016). Importantly though, with the sensitivity of fMRI, we did not find enhanced responses in fronto-parietal cortex, nor in subcortical regions such as the colliculus superior and typical valence-related regions such as the nucleus accumbens and orbitofrontal cortex, as expected if attention was cued toward the paired cue. Therefore, local changes in inferotemporal cortex provide the most parsimonious explanation for the behavioral and functional changes we observed.”

2) It’s a bit inelegant to have different tasks for measuring cortical responses in Experiment 1 and reading out behavioral changes in Experiment 2. This considerably weakens the authors’ attempt at providing the first information about perceptual learning and cortical plasticity with VTA-EM in primates, as the learning and plasticity relate to fundamentally different kinds of stimuli. (And thus there might be substantially different representational changes in Experiment 2 that provide the behavioral change; conversely there may be no consequence of the changes measured in Experiment 1.)

We intentionally chose to start with a grating experiment followed by the motion experiment. After the grating experiment (experiment 1), we were not sure that we would have changed the network already in such a manner that it would prevent detecting behavioral effects induced by VTA-EM. Therefore, we decided to use the coherent motion stimulus presented at another location in the visual field instead of gratings. Despite this, we agree with the reviewer that demonstrating physiological plasticity utilizing the same stimuli as the behavioral changes would be a more elegant demonstration of the effects of pairing with VTA-EM.

Therefore, in an effort to determine where changes occur after Pavlovian association of VTA-EM with motion stimuli, we have added experiment 3 in which we also acquired fMRI responses to the motion stimuli before and after a cue-VTA-EM association phase. Like the fMRI data acquired in Experiment 1, the animals performed the orthogonal grating task concurrently while the stimuli of interest (motion stimuli, 2% coherence, upward or downward, LVF or RVF) was displayed. Intriguingly, we found enhanced responses for the paired motion direction when presented in the paired visual field. These enhancements were also restricted to the hemisphere contralateral to the paired visual field in an inferotemporal region near PIT and, surprisingly, not in typical motion-processing areas such as MT (see figure below). This activation was located at the same position along the anterior-posterior axis as PIT but, surprisingly, located more ventro-medially. This evidence further supports the proposition that, at least for simple orientation and motion stimuli, pairing with VTA-EM in a task-irrelevant manner specifically enhanced by stimulus representations in posterior regions of inferotemporal cortex.

Manuscript New Figure

Figure 5. Experiment 3: Responses to weak motion stimuli selectively enhanced after VTA-EM. Flat maps depict main effect of pairing (second-level linear model, main effect of pairing – pre- vs. post-association, $p < 0.001$, cluster size 30 mm^2 , M1 – 77 runs pre, 48 runs post; M2 – 47 runs pre, 43 runs post, see Methods) on the direction response (paired vs. control direction) for 2% coherent motion for presentations to **a** control VF and **b** paired VF projected onto a flattened cortical representation. Black outlines depict visual areas defined by retinotopic mapping (Janssens et al 2014).

3) What makes dorsal V3 and PITv particularly sensitive to VTA-EM?

We can only speculate about this question but we hypothesize that it is the interaction of weak, stimulus driven activity and VTA innervation that results in preferentially stronger plasticity we observed in posterior IT regions. More specifically, we hypothesize that the low contrast, low SNR/coherence and simple stimuli (grating/motion stimuli) weakly activated visual cortical regions from earlier visual areas to the more posterior parts of the temporal lobe. In contrast, dopamine receptor and retrograde tracer studies indicate that dopaminergic VTA innervation is increasingly stronger as one moves from posterior visual cortex (V1) to anterior temporal regions (Berger et al 1991, Gattass et al 2014). Therefore, based on the activity elicited by the stimuli and dopaminergic input to visual regions, we feel that mid-tier visual regions receive both the necessary visual activity and the VTA-mediated dopaminergic input that would be needed to drive plasticity. We have made this hypothesis more explicit in the discussion (see below). Also note that we do not dispute in any manner that no plasticity can happen earlier or later in the system. Our evidence points toward the strongest effects in mid-level areas, at least with the experimental paradigm we used.

“The enhanced response to the paired stimulus (based on general linear model analyses) and improved orientation classification (based on searchlight MVPA) was strongest in posterior inferotemporal (IT) areas. This indicates that posterior IT, and mid-stage visual areas more generally, may have a higher susceptibility to the VTA-driven component of VPL-related plasticity. ... The stronger plasticity observed within these visual regions found with VTA stimulation and reinforcement learning may reflect the interaction of stimulus-driven activity and dopaminergic innervation within mid-stage regions. In addition, this hypothesis suggests that weak, unattended stimuli reduce the activity needed to generate plasticity in higher visual areas while sparser dopaminergic innervation in earlier visual regions renders these regions less conducive to plasticity. This is supported by studies of primate dopamine receptor distribution that have found more pronounced receptor densities in temporal compared to primary visual cortex (Berger et al 1991). In contrast, other neuromodulatory systems with stronger innervation

to early visual regions (Kosofsky et al 1984) may also play a role in VPL related plasticity observed in these earlier regions (van Kerkoerle et al 2018, Yamahachi et al 2009). The latter hypothesis requires further testing.”

4) Does VTA-EM utilize dopamine, and/or co-release of some sort of factor to lead to these improvements?

In collaboration with colleagues Christin Sander, Bruce R. Rosen and Joe B. Mandeville we have been examining dopamine release induced by VTA-EM in anesthetized baboons employing the same microelectrode bundle used in the current study. We measured the dopaminergic contribution of fMRI responses to VTA-EM during pharmacological challenges with a D1 antagonist (SCH-23390 @ 0.03 – 0.09 mg/kg) and a D2 antagonist (Prochlorperazine @ 0.1 mg/kg). The doses of D1 and D2 antagonist used had previously been shown to achieve ~90% occupancy. The VTA-EM stimulation paradigm, used in the baboon study, was very similar to that used in the current study with the same duration (200 ms) and frequency of stimulation (100 Hz). The anesthetized baboon study used a larger current (1mA) than our awake rhesus monkey study (150 μ A – 400 μ A). Note, however, that a previous fMRI study found similar activations between anesthetized and awake experiments when awake currents were ~2-5 times larger (Premereur et al 2015).

Measurement of the fMRI activity evoked by VTA-EM in baboons revealed that D2 blockade enhanced fMRI activity while D1 blockade reduced activity (See Figure 1 below). From these results it was calculated that ~31% of fMRI signal in putamen was D2R mediated while ~38% was D1R mediated with 31% hypothesized to be non-dopaminergic (Sander et al 2018). Therefore, VTA-EM in primates, utilizing highly similar stimulation parameters, was found to robustly elicit dopamine dependent modulation of the caudate, putamen and nucleus accumbens.

Figure of VTA-EM evoked fMRI activity under pharmacological D1 and D2 blockade.

Based on this evidence it is highly likely that VTA-EM during our study triggered dopamine release. Nonetheless, the above-mentioned fMRI results may indicate that a non-negligible portion of the VTA-EM driven activity is likely non-dopaminergic. It should be noted that a population of dopaminergic VTA neurons also co-release GABA and glutamate (Berrios et al 2016, Morales & Margolis 2017, Zhang et al 2015) so even optogenetic stimulation of VTA neurons would not definitively implicate dopaminergic signaling in observed effects. Therefore, we can only causally link VTA activation with the observed physiological and behavioral effects while the role of dopaminergic signaling is correlative. We have made sure to more explicitly discuss this distinction in the discussion (see below).

In addition, we are currently piloting dopamine specific optogenetic stimulation in primates. However, although we use protocols that appeared to be successful in prior studies (Stauffer et al 2016), we encountered several difficulties with dopaminergic cell-type specific optogenetics in primates. Of considerable concern has been a lack of expression, potentially resulting from a weak TH promoter, and a surprising non-specificity of reportedly selective constructs. Despite these difficulties, we do feel that optogenetic manipulations hold best chance to unravel the mechanism behind VTA-driven visual plasticity in primates.

“Therefore, despite earlier work demonstrating that primate VTA-EM elicits dopamine release (Sander et al 2018, Schluter et al 2014), the mechanism through which primate VTA-EM affects behavior and physiology and whether that is dependent on dopamine alone remains to be determined.”

5) Any chance this involves fibers of passage along with or instead of direct VTA stimulation?

Evidence does suggest that axons passing the electrode tip can be excited by microstimulation. Therefore, axons which are only passing through the VTA and are in close proximity to the stimulating electrode could also be activated. Nonetheless, the activation of fibers of passage likely has a negligible effect on regions outside the VTA. Two photon imaging of the cells stimulated by microstimulation have found most activated voxels are in close proximity of the electrode although some activated cells were found millimeters away (Histed et al 2009). This is also supported by behavioral evidence from microstimulation. Histed et al (2013) explained that “Such selective behavioral effects likely reflect the concentration of activated neurons immediately around the stimulating electrode tip. Because postsynaptic effects typically require the summation of many inputs, the downstream effects of stimulation might depend on relatively closely-spaced activated neurons immediately around the electrode tip. The widely spaced activated cells far from the electrode might be sufficiently sparse that they do not converge on the postsynaptic neurons.” The sparsity of activations farther from the electrode tip has been investigated on a larger spatial scale using fMRI. In V1 the relationship between stimulation current and activated voxels was described by the equation $I = Kr^2$ where I = current in μA , r = radius in mm and $K = 1292 \mu\text{A}/\text{mm}^2$ (Tolias et al 2005). Based on these findings we would expect a radius of activation (0.34 – 0.56 mm) or a volume of activation (0.17 - 0.72 mm^3) for our stimulation using currents (150 μA – 400 μA). Finally, our own study examining the cortical activations elicited by VTA-EM found that the network of structures activated by VTA-EM was largely

comprised of regions with dense dopaminergic connections indicating that stimulation is relatively confined to the VTA (see below figure 4 adapted from Arsenault et al 2014). Finally, comparisons from the lab of optogenetics and microstimulation (see below Figure 3 adapted from Gerits et al 2013) revealed a highly similar network of structures activated by these two techniques. Moreover, both optogenetics and electrical microstimulation revealed a pattern of activations highly similar to connectivity patterns obtained using conventional anatomical tract tracing methods. Again, this suggests that microstimulation predominantly affects neurons in close proximity to the electrode. Therefore, we feel that the effects we measured after VTA-EM largely result from the local activation of VTA.

Adapted from Figure 4 of (Arsenault et al 2014). Group analysis T score maps overlaid on coronal slices of the 112 RM-SL T1/T2* anatomical volume (n = 35 runs, M1 = 12 runs, M2 = 5 runs, M3 = 18 runs, fixed effect analysis, VTA-EM – no VTA-EM, FDR corrected, $p = 0.001$, cluster size: 10 voxels). VTA-EM consisted of a 200 ms train of bipolar stimulation pulses (200 Hz; 200 ms; 100 μA –392 μA ; two VTA electrodes stimulated simultaneously).

The following abbreviations were used: AIP, anterior intraparietal; cnMD, centromedian nucleus; Cd, caudate; DO, dorsal opercular; G, gustatory; GrF, granular frontal; Hc, hippocampus; NA, nucleus accumbens; PAG, periaqueductal gray; Pu, putamen; PrCo,

precentral opercular; RN, red nucleus; TPO, temporal parietal occipital; VL, ventral lateral nucleus.

Adapted from Figure 3 of (Gerits & Vanduffel 2013). Comparison between (A) anatomical tractography, (B) electrical microstimulation (EM), and (C) optical stimulation. Injections of tracer in the frontal eye fields (FEF) of macaque monkey resulted in labeled cells in the lateral intraparietal area (LIP), the medial superior temporal area (MST), and the superior temporal polysensory area (STP)(Huerta et al 1987, Schall et al 1995). Functional MRI (fMRI) combined with EM of the FEF resulted in fMRI activations in LIP, MST, and STP (Ekstrom et al 2008). Monkey opto-fMRI with ChR2-transduced neurons in FEF also showed an increase in fMRI signal in LIP, MST, and STP (Gerits et al 2012)[12]. Reprinted, with permission, from (Gerits et al 2012)(C), (Schall et al 1995) (A), and (Ekstrom et al 2008)(B).

6) Studies of neuromodulation and plasticity generally indicate that the details of the stimulation association are critical in determining outcomes, perhaps especially so for VTA activation. The authors should clarify in the main text or figures how the stimulus was performed relative to visual stimulus presentation.

Note: We believe the question “how the stimulus was performed relative to visual stimulus presentation” should instead be “how the stimulation was performed relative to visual stimulus presentation”. Therefore, we will answer the question based upon this interpretation.

*See answer to Reviewer 1 Question 7 for a discussion of the rationale for our choice of timing.

We initially described the temporal interval between stimulus presentation and VTA-EM in Experiments 1 & 2 using only the figures and the figure legend. More specifically, in experiment 1 we described this relationship with Figure 2B (see below). The figure legend for figure 2B explained “During cue-VTA-EM association training, after a 300 ms delay, the L45° stimulus was coupled with VTA-EM (200 ms, 100Hz, material and methods).”

Figure 2B from manuscript.

We also described the timing of stimulation in experiment 2 using Figure 4D. The figure legend of Figure 4D explains “Only one condition (e.g. 2% left upward) was paired with VTA-EM (300 ms delay, 200 ms, 100Hz, methods).”

Figure 4D from paper.

We thank the reviewer for this comment and acknowledge that our original explanation of the timing of microstimulation was not sufficient. Therefore, to help clarify and more directly address the importance of the timing of VTA-EM and stimulus pairing we have now added an additional explanation of this relationship to the main text and figure legends. In addition, we also adapted the figures to improve the visualization of the timing details.

Experiment 1

Adapted Text:

“Because evidence suggests that delay conditioning creates relatively stronger associations between unconditioned and conditioned stimuli (Bao et al 2003, Kryukov 2012, Werden & Ross 1972), the 200 ms train of VTA-EM began 300 ms after the onset of the grating stimulus (Fig. 2b). Therefore, the conditioned stimulus (i.e. grating stimulus) preceded and overlapped with the unconditioned stimulus (i.e. VTA-EM).”

Adapted Figure 2B

Adapted Figure 2B Legend

“During cue-VTA-EM association, the L45° stimulus was coupled with VTA-EM (100Hz, see Methods). VTA-EM (200 ms duration) began 300 ms into the stimulus presentation (500 ms duration).”

Experiment 2

Adapted Main Text

“A delay forward conditioning paradigm was also employed in experiment 2, therefore VTA-EM (200 ms duration) began 300 ms after the onset of the paired motion stimulus (500 ms duration).”

Adapted Figure 4D

7) How did the authors choose their parameters for association and VTA-EM? What would happen if the temporal interval between VTA-EM and visual stimulus was altered?

Based on our results described in Arsenault et al. (2014), we have a strong hypothesis that associations between VTA-EM and a visual stimulus behave like an unconditioned (US) and conditioned stimulus (CS) in Pavlovian conditioning paradigms. Therefore, we chose forward conditioning [US (VTA-EM) starting **after** the CS (visual stimulus) onset] over backward conditioning [US (VTA-EM) starting **before** the CS (visual stimulus) onset] because:

1) backward conditioning can inhibit association learning (Siegel & Domjan 1974).

2) backward conditioning, with VTA-EM as the US, was shown to reduce cortical representations of CS while forward conditioning was shown to increase the cortical representation of the CS (Bao et al 2003).

In addition, we also chose delay [temporal overlapping of CS (visual stimulus) and US (VTA-EM)] over trace [a temporal delay between CS (visual stimulus) and US (VTA-EM)] forward conditioning. Delay conditioning was chosen because delay conditioning creates stronger associations than trace conditioning (Kryukov 2012). Therefore, VTA-EM (US) began 300 ms after visual stimulus (CS) onset as a confluence of earlier decisions to **1)** use delay conditioning, **2)** 200 ms of VTA-EM and **3)** 500 ms visual stimulus.

As the reviewer suggested, we do expect that the timing of VTA-EM relative to the visual stimulus would affect the physiological and behavioral changes we monitored. The existing literature indicates a continuum moving from **A)** backward conditioning (CS before US) == reversed or inhibited learning/plasticity to **B)** delay forward conditioning (CS overlaps with US) == high learning/plasticity to **C)** trace forward conditioning (CS after US) == lower learning/plasticity). Based on this it would be highly interesting to

examine how timing of the visual stimulus relative to VTA-EM affects learning/plasticity in our task-irrelevant learning paradigm. Despite this, the experiments undertaken in this study were extremely time-consuming. An average round of experiment 1 & 2 lasted for 16.5 x 3-4 hr. daily sessions or 22.5 days if weekends are included. Therefore, due to the time-consuming nature of these experiments we could not allocate the time needed to examine the admittedly interesting effects of cue-VTA-EM timing. Instead we focused on delay conditioning because we felt we would have the best chance to determine whether VTA-EM could generate task-irrelevant visual perceptual learning. To better address the rationale for timing the text below has been added to the results section.

“Because evidence suggests that delay conditioning creates relatively stronger associations between unconditioned and conditioned stimuli (Bao et al 2003, Kryukov 2012, Werden & Ross 1972), the 200 ms train of VTA-EM began 300 ms after the onset of the grating stimulus (Fig. 2b). Therefore, the conditioned stimulus (i.e. grating stimulus) preceded and overlapped with the unconditioned stimulus (i.e. VTA-EM).”

8) Were the multiunit recording studies in Figure 1 done for both monkeys, or just one? Either way this should be clarified; if both, the data shown separately for each.

The data from a single animal are shown. We have clarified this in the text, and methods. The other animal is currently involved in a prolonged series of time-sensitive experiments. Therefore, acquiring additional electrophysiological data from this animal would greatly delay the resubmission of this manuscript. Meanwhile, we have collected single unit responses using acute VTA recordings from a third animal in the context of an entirely different series of experiments but using the same reward-prediction error paradigm as for the multiunit recordings of the present experiments. VTA-targeting procedures in this animal were identical and the single unit responses were also highly similar to those of the present study.

Adapted Text

“To confirm the accuracy of the electrode positioning procedure, we recorded multi-unit activity (MUA) from subject M1 during a classical conditioning paradigm.”

Adapted Figure Legend

“Visual stimuli were presented to subject M1 for 500 ms and associated with a unique reward probability (0%, 25%, 50%, 75%, 100%).”

9) It's not clear how the authors are using the word 'training'. I'd encourage them to strictly use this term to refer to behavioral training, and use some other term to refer to the VTA-EM procedure.

We chose the term ‘training’ to refer to the repeated association of a visual cue with VTA-EM because several task-irrelevant visual perceptual learning papers, published in multi-disciplinary journals, have used training in a similar way (Amano et al 2016, Seitz et al 2009, Shibata et al 2011). In the paper we tried to clarify this at the end of the last paragraph of the introduction.

Original manuscript

‘Importantly due to the task-irrelevant experimental design, “training” in this paper does not refer to explicit task performance but instead the passive association of an unattended visual stimulus with VTA-EM.’

Nonetheless, we agree with the reviewer that the passive “training” we referred to in the paper could easily be confused with explicit behavioral training on a task. Therefore, throughout the manuscript we have replaced “trained” with “paired” to refer to the stimulus paired with VTA-EM. We feel this change will help clarify the experimental design for readers and avoid unnecessary confusion.

10) Effect size of VTA-EM is small, a change of d' of about 0.3 (or less than 10%).

The effect size is small but consistent across animals and rounds of experiment 2. This consistency can be seen in supplemental Figure S4A in which a relative improvement in performance for the paired visual field is found for all rounds. This consistency is also supported by the analysis of d' -prime data in which we find a significant effect of stimulation after controlling for monkey and round (see response to reviewer 2 major question 3). Despite the reliability of the observed learning we agree with the reviewer that the observed improvement is not exceedingly large. We feel that the magnitude of the observed effect may be the result of a few, distinct factors. One is that the animals were already well trained on the task. Therefore, there is a ceiling to how well the monkeys can perform with a given amount of motion information. Ideally, we would run this experiment in naive animals but because significant training is required to be sure the animals “understand” the task this is not trivial. Another important consideration is that the task-irrelevant, Pavlovian component of perceptual learning is only a portion of a typical perceptual learning process. Under normal perceptual learning, stimulus-directed attention and task-oriented behavior interact with the associative process. Importantly, we are only trying to demonstrate the role of the primate VTA in task-irrelevant learning. We strongly believe that combining attentional selection with VTA stimulation would result in stronger effects. The problem with such a design would be that possible physiological and perceptual effects could be entirely attributed to attention, unlike the what we show in the present study. In addition, as suggested by the reviewers the timing and frequency of VTA-EM is likely not optimized for maximal learning. Unfortunately, the time-consuming nature of these experiments precluded a detailed study of VTA-EM parameter space. Therefore, we were restricted to selecting VTA-EM parameters, that based on literature and prior experience (see Reviewer 1 Question 7), provided the best chance for success. Finally, our reply to Reviewer 1 Question 11, also adds to the explanation of the relatively small behavioral

effects. Importantly, we have clarified that the observed changes will, by no means, account fully for those observed in normal perceptual learning.

“It is also important to note that this Pavlovian component of perceptual learning that we examined using VTA-EM is only part of a typical perceptual learning process. Consequently, the small but consistent improvements we observed in d-prime are likely weaker than would be observed with a task-relevant design. Therefore, although there is a large degree of similarity between task-driven perceptual and procedural learning (Censor et al 2012) and a likely overlap exists between the mechanisms generating these types of learning, our work adds to a growing body of evidence demonstrating a Pavlovian neuromodulatory component to perceptual learning.”

11) Why does discrimination of stimuli in the control VF get worse by about the same magnitude as the ‘trained’ VF improves?

Pairing (i.e. association of a visual cue with VTA-EM) during experiment 2 was performed for 6 - 20 sessions per round (mean # of sessions = 11.5 sessions, mean # of days between sessions with weekend = 15.5 days). Therefore, the monkeys had not performed the motion discrimination task for the duration of cue-VTA-EM association sessions plus weekends that overlapped with the training sessions. A lack of recent task exposure tends to reduce task performance for monkeys. This is especially pronounced for difficult tasks. Therefore, it is not surprising that task performance was reduced in the control VF because of the long duration of time since the motion discrimination task was last performed. This underlines the importance of the control VF for comparison when the monkey’s performance could change due to factors not related to the intervention. In essence, this also means that we are probably underestimating the true perceptual benefits induced by VTA-EM as we do not take into account the ‘unlearning’ or extinction that must have occurred in the trained VF.

“By taking the control VF into account, we control for possible time-dependent reductions in performance resulting from the time between the pre- and post-association phases. ... It is important to note that the monkeys did not perform the motion discrimination task for the duration of cue-VTA-EM association sessions plus weekend days that overlapped with these association sessions. A lack of recent task exposure tends to reduce task performance of monkeys, especially for difficult tasks. Therefore, it is not surprising that task performance was reduced in the control VF because of the long duration of time since the motion discrimination task was last performed. This also means that we are probably underestimating the true perceptual benefits induced by VTA-EM as we do not take into account the ‘unlearning’ that must have occurred in the paired VF.”

12) There are citations missing that provide critical background and context for these studies. The author are wrong in their central claims in the abstract and introduction that: ‘no experimental evidence exists confirming the capacity of neuromodulatory regions to cause VPL and related adult cortical plasticity in primates’ and ‘rodent studies have elegantly demonstrated stimulus-specific plasticity in sensory cortex driven by activation of neuromodulatory centers (Bao et al., 2001; Liu et al., 2015), but these studies have not examined this with respect to perceptual learning nor selective attention’. First note that those two references are not numbered and in the main reference list. Perhaps there’s no one specific paper looking at visual perceptual learning and VTA stimulation with monitoring of cortical responses, but the authors’ claim is still overblown in terms of the implications. I suggest the authors take a look and think about integrating other past work in primates (Liu et al. Current Biology 2017) and rodents (Froemke et al. Nature Neuroscience 2013; Livneh et al. Nature 2017; Ogg et al. Nature Communications 2018; Pinto et al. Nature Neuroscience 2013). I would say there’s growing evidence connecting sensory cortical changes to specific enhancements in sensory processing and perceptual learning in different species.

The first reviewer's concerns are echoed by reviewer 2. In an effort to keep the manuscript condensed we regrettably omitted a larger discussion of the body of surrounding work. We have significantly expanded the discussion of the cortical plasticity in sensory regions especially focusing on the substantial work done in the auditory system. In doing this we feel we have greatly improved the clarity of our novelty claims.

Nonetheless, with reference to our claim that “no experimental evidence exists confirming the capacity of neuromodulatory regions to cause VPL and related adult cortical plasticity in primates” we feel this statement does not overreach. The primate study cited by the reviewer examines the effects of nucleus basalis on working memory (i.e. performance on delayed match-to-sample task with or without nucleus basalis stimulation turned on). This study does not examine perceptual learning nor cortical plasticity (the only neural responses assessed were immediate changes in LFP at the NB stimulating site). Based on this, we feel our statement is highly focused on the role of neuromodulatory regions in visual perceptual learning and should not lead readers to incorrectly assume the function of neuromodulatory regions is not being investigated in primates.

In contrast and much in agreement with the comments of reviewer 1 here and reviewer 2, our discussion of the rodent work was under-represented and should be expanded accordingly. Therefore, we have expanded this discussion and amended our insufficient claims about this work (see below). We thank the reviewers for guiding us in this direction.

Amended text (introduction)

“Rodent studies have elegantly demonstrated stimulus-specific plasticity in visual cortex driven by activation of neuromodulatory centers (Chubykin et al 2013, Liu et al 2015), but these studies have not examined this with respect to perceptual learning nor selective attention.”

Amended text (discussion)

“Comparison with neuromodulatory-driven sensory plasticity and perceptual learning in rodents

Pioneering rodent work has demonstrated that pairing stimulation of both dopaminergic and cholinergic neuromodulatory centers, with simple auditory stimuli increases stimulus representations in auditory cortex (Bao et al 2001, Kilgard & Merzenich 1998). Moreover, these representational enhancements coincide with perceptual learning (Froemke et al 2013). Further work has shown, that while increased stimulus representations correlate with improved stimulus perception, these enhancements dissipate over time, unlike the behavioral improvements (Reed et al 2011). This indicates that largescale representational enhancements, like those observed in the current study, may be a transient stage in the learning process that is later pruned into sparser representational enhancements. In reference to task irrelevant perceptual learning, the common use of anesthesia during rodent studies while pairing nucleus basalis stimulation (Froemke et al 2013, Reed et al 2011) has provided perhaps the most compelling proof that Pavlovian association, in the absence of attention, can generate perceptual learning. On the other hand, anesthesia may also interact with the normal functioning of the circuitry involved in different types of learning.

In the visual domain, long-term changes in rodent primary visual cortical responses have been found after reward pairing (Shuler & Bear 2006). Further work has causally demonstrated that direct cholinergic input to area V1 drives the acquisition of these responses (Chubykin et al 2013, Liu et al 2015). In addition to these longer-term changes, fast improvements in visual perception have been observed when cholinergic activity is acutely increased (Pinto et al 2013). This indicates that the influence of neuromodulators on perceptual learning occurs across many timescales.

Across rodent studies examining the role of neuromodulatory regions, some of the most powerful designs utilize transgenic lines to target specific cell types. In contrast, considerable obstacles exist for cell-type specific modulations in primates (but see (Stauffer et al 2016)). Therefore, despite earlier work demonstrating that primate VTA-EM elicits dopamine release (Sander et al 2018, Schluter et al 2014),

the mechanism through which primate VTA-EM affects behavior and physiology and whether that is dependent on dopamine alone remains to be determined. Nonetheless, important species differences, like reduced visual acuity in rodents (Prusky & Douglas 2004) and stronger cortical dopaminergic innervation observed in primates compared to rodents (Berger et al 1991) suggests the mechanisms underlying VPL across species may vary substantially.”

Reviewer #2 (Remarks to the Author):

Arsenault and Vanduffel have designed two experiments that allow to clarify an important question, namely whether visual perceptual learning is driven by reinforcement and corresponding neuromodulatory centers or by attention-related mechanisms enhancing stimulus-driven activity. To this end they classically condition task-irrelevant stimuli with reinforcement provided by electrically activating the dopaminergic midbrain and examine whether this treatment results in changes in visual cortex fMRI activation and in improved perception tested in an operant task.

The first experiment consists of three phases. In the first phase they measure fMRI activation to task-irrelevant grating stimuli presented at different positions in the visual field while monkeys perform a visual task on different stimuli presented at other positions of the visual field. Subsequently one of the irrelevant grating stimuli was repeatedly paired with dopaminergic midbrain stimulation. This second phase was followed by a third phase repeating phase 1 and revealed a selective enhancement of the fMRI activation in the PITv to the irrelevant grating stimulus paired in phase 2. An additional searchlight analysis confirmed that only during phase 3 activity patterns in PITv contained sufficient information to classify the orientation of the grating stimuli.

The second experiment also consisted of three phases. In the first phase the monkeys performed a task different from that in experiment 1 and with different stimuli placed at different locations of the visual field. In phase 2, the monkeys performed the visual task of Experiment 1 while one the stimuli used in the task of phase 1 of Experiment 2 was repeatedly paired with dopaminergic midbrain stimulation. This treatment resulted, in phase 3, in enhanced performance of the task of phase 1 for the stimulus selected for classical conditioning in phase 2.

The experiments are very well designed and very well conducted, and the data are analysed with the state-of-the-art methods and using appropriate statistical tests. In general this research is highly relevant to a wider audience and the results

are sound. At this stage, however, my enthusiasm about this research is somewhat dampened by the two issues detailed below.

Major

1) The information given in the manuscript is highly compact. This makes it difficult for scientists not experts of visual perceptual learning to understand the manuscript. If the authors aim to publish their work in a non-specialist journal they should try to do a better job in describing their findings to a wider audience.

We thank the reviewer for this helpful comment and we agree that the original manuscript was overly compact and this led to a lack of clarity for non-experts in visual perceptual learning. We have made changes throughout the text to clarify specialized terms and expand on condensed explanations. In addition, we had colleagues review the manuscript for regions that were unclear and we addressed their concerns. It is difficult to pick a section of text to exemplify these changes, but we hope that the revised form of the manuscript clarifies our design, analysis and results for a wider audience.

2) In addition, they should relate their findings to the rich literature on auditory perceptual learning. Here, for example, it was shown that induction of cortical map plasticity by stimulation of the nucleus basalis improves learning (Reed A et al., *Neuron*. 2011 Apr 14;70(1):121-31; for review see also Irvine, D.R.F, *Hearing Research* 366, September 2018, Pages 3-16). They should specify how their work on visual perceptual learning advances over what is known on auditory perceptual learning and on procedural learning (Censor N et al. 2012. *Nat. Rev. Neurosci.* 13, 658e664).

Due to our overly compact style of our original submission we failed to properly address the work that our findings build upon. We have expanded our discussion accordingly. Some examples of this expansion can be seen in below:

Amended text (introduction)

“Rodent studies have elegantly demonstrated stimulus-specific plasticity in visual cortex driven by activation of neuromodulatory centers (Chubykin et al 2013, Liu et al 2015), but these studies have not examined this with respect to perceptual learning nor selective attention.”

Amended text (discussion)

“Comparison with neuromodulatory-driven sensory plasticity and perceptual learning in rodents

Pioneering rodent work has demonstrated that pairing stimulation of both dopaminergic and cholinergic neuromodulatory centers, with simple auditory stimuli increases stimulus representations in auditory cortex (Bao et al 2001, Kilgard & Merzenich 1998). Moreover, these representational enhancements coincide with perceptual learning (Froemke et al 2013). Further work has shown, that while increased stimulus representations correlate with improved stimulus perception, these enhancements dissipate over time, unlike the behavioral improvements (Reed et al 2011). This indicates that largescale representational enhancements, like those observed in the current study, may be a transient stage in the learning process that is later pruned into sparser representational enhancements. In reference to task irrelevant perceptual learning, the common use of anesthesia during rodent studies while pairing nucleus basalis stimulation (Froemke et al 2013, Reed et al 2011) has provided perhaps the most compelling proof that Pavlovian association, in the absence of attention, can generate perceptual learning. On the other hand, anesthesia may also interact with the normal functioning of the circuitry involved in different types of learning.

In the visual domain, long-term changes in rodent primary visual cortical responses have been found after reward pairing (Shuler & Bear 2006). Further work has causally demonstrated that direct cholinergic input to area V1 drives the acquisition of these responses (Chubykin et al 2013, Liu et al 2015). In addition to

these longer-term changes, fast improvements in visual perception have been observed when cholinergic activity is acutely increased (Pinto et al 2013). This indicates that the influence of neuromodulators on perceptual learning occurs across many timescales.

Across rodent studies examining the role of neuromodulatory regions, some of the most powerful designs utilize transgenic lines to target specific cell types. In contrast, considerable obstacles exist for cell-type specific modulations in primates (but see (Stauffer et al 2016)). Therefore, despite earlier work demonstrating that primate VTA-EM elicits dopamine release (Sander et al 2018, Schluter et al 2014), the mechanism through which primate VTA-EM affects behavior and physiology and whether that is dependent on dopamine alone remains to be determined. Nonetheless, important species differences, like reduced visual acuity in rodents (Prusky & Douglas 2004) and stronger cortical dopaminergic innervation observed in primates compared to rodents (Berger et al 1991) suggests the mechanisms underlying VPL across species may vary substantially.

...

Therefore, although there is a large degree of similarities between task-driven perceptual and procedural learning (Censor et al 2012) and a likely overlap exists between the mechanisms generating these types of learning, our work adds to a growing body of studies demonstrating a Pavlovian neuromodulatory component to perceptual learning.”

3) It did not become clear to me how Experiment 2 was analysed. Unfortunately, the information given in the Legend of Figure 4 is very sparse ('40 bins/round, 100 trials/bin...') and the Methods are not fully helpful. Which trials and which sessions were used to determine the 'pre' and the 'post' motion discrimination sensitivity of the nonhuman primates? Did the authors combine the performance of Monkey 1 and Monkey 2? Did the authors average all pre-conditioning trials and compare them with the average of all post-conditioning trials? If this is true I wonder

whether their analysis can contribute to the question whether their conditioning paradigm resulted in 'learning'.

We agree with the reviewer that the explanation of the analysis of experiment was not clear enough. We have attempted to make our analysis clearer by expanding our discussion of data partitioning and data analysis within the Methods. In addition, based on the reviewers comments we have changed our statistical analysis of the behavioral data to a linear mixed effects (LME) model. We think that the LME model more effectively controls for inter-subject/round variability and thank the reviewer for guiding us to this improvement. Consequently, in the amended manuscript, we include the more detailed explanation of the data partitioning in the methods section and the original ANOVA model is replaced with the better controlled LME model. In addition, a table (Point to Point Table 1) is provided below for the reviewers that details the data partitioning.

Methods

Experiment 2: d-prime and c-criterion calculation. Hit rate and false-alarm rate were calculated separately for the paired VF and the control VF. Hits were defined as a correct response for presentations of the paired direction while false-alarms denoted an incorrect response for the control direction. Using these definitions, d-prime and c-criterion were calculated separately for bins of 100 trials. D-prime was calculated as the difference between z-scores of the hit rate and the false-alarm rates. C-criterion was calculated as the average of the z-scores of the hit rate and the false-alarm rates multiplied by -1. The c-criterion is an indicator of the bias to report the paired direction with more negative values indicating a stronger bias.

Experiment 2: Partitioning of behavioral data (general). For each pre-test (n=6 rounds, 2 monkeys x 3 rounds/monkey) the d-prime and c-criterion for each coherence level was calculated for 40 bins, each bin containing 100 trials. Therefore, for a given round, the 40 data points in the pre-association bins were taken from the 4,000 trials occurring directly before the association round. For each post-test (n=6,

3 rounds x 2 monkeys) the d-prime and c-criterion for each coherence level was calculated for 20 bins of 100 trials. Therefore, for a given round, the 20 data points in the post-association bins were taken from the 2,000 trials directly after the association round –we took 2,000 trials to estimate changes in performance directly after the association phase. From this partitioning, each round performed by each monkey (n=6, 2 monkeys x 3 rounds/monkey) had 40 data points to describe the pre-association c-criterion or d-prime of a particular coherence level and 20 data points to describe the post-association c-criterion or d-prime of a particular coherence level.

Experiment 2: Partitioning of behavioral data (d-prime). From the general partitioning of data described above, we examined changes resulting from VTA-EM association for parathreshold levels of motion coherence. Parathreshold coherence levels of 12% and the 6% motion coherence level were utilized for M1 and M2, respectively. These motion coherences represented the lowest signal strength above 0% coherence levels where comparable pre-association motion discrimination sensitivity was found for both monkeys (mean pre-association d-prime value between 3-4).

Experiment 2: Analysis of behavioral data (d-prime). We constructed hierarchical mixed effects models(Bates et al 2015) with monkey and the rounds each monkey performed as random effects factors and visual field (control or paired) and pairing (pre or post) as fixed effects factors. We compared a random slopes model to a random intercepts model using the Akaike information criterion (AIC) for model selection:

Interaction of visual field and pairing

random slopes model: $F(1,14.098)=4.679$, $p=0.0482$, AIC = 3257.83

random intercepts model: $F(1,698)=5.512$, $p = 0.01917$, AIC = 3249.44

AIC values were lower for the random intercept model compared to the random slopes model, indicating the random intercept model was the most parsimonious.

Partitioning of data (c-criterion)

For the analysis of changes in c-criterion after VTA-EM, we examined all stimulus strengths except for the 50% motion coherence. 50% motion coherence was excluded because animals performed close to perfectly at this coherence level leaving little room for changes in bias due to VTA-EM.

Analysis of data (c-criterion)

We constructed hierarchical mixed effects models(Bates et al 2015) with monkey, rounds each monkey performed, and the snr levels displayed in each round as random effects factors and visual field (control or paired) and pairing (pre or post) as fixed effects factors. We compared a random slopes model to a random intercepts model using the AIC for model selection:

Fixed Effects: Interaction of visual field (control or paired) and pairing (pre or post)

random slopes model: $F(1,47.402)=16.688$, $p=0.000169$, AIC = 11862.86

random intercepts model: $F(1,3462.8)=22.954$, $p = 1.73 \times 10^{-6}$, AIC = 11911.05

AIC values were lower for the random slopes model compared to the random intercept model, indicating the random slopes model is the most parsimonious model.

monkey	round	paired VF	pre-post	# of data points (D-prime from bins of 100 trials)
M1	1	Paired	Pre	40
M1	2	Paired	Pre	40
M1	3	Paired	Pre	40
M1	1	Control	Pre	40
M1	2	Control	Pre	40
M1	3	Control	Pre	40
M1	1	Paired	Post	20
M1	2	Paired	Post	20
M1	3	Paired	Post	20
M1	1	Control	Post	20
M1	2	Control	Post	20
M1	3	Control	Post	20
M2	1	Paired	Pre	40
M2	2	Paired	Pre	40
M2	3	Paired	Pre	40
M2	1	Control	Pre	40
M2	2	Control	Pre	40
M2	3	Control	Pre	40
M2	1	Paired	Post	20
M2	2	Paired	Post	20
M2	3	Paired	Post	20
M2	1	Control	Post	20
M2	2	Control	Post	20
M2	3	Control	Post	20

Point to Point Table 1. Provides an overview of the data points used in the d-prime ANOVA analysis

Minor

1) There are some instances where the manuscript is poorly edited. For example, on page 3 references are not indicated by numbers. On page 4, the name ‘Kruskal’ is spelled in three different ways, the names ‘Bonferroni’ and ‘Wallis’ should also be corrected.

These typos have been fixed. We apologize for these mistakes.

2) On page 3, please explain in which way is the second task ‘orthogonal’ to the first task? Do you mean the two tasks are independent from each other?

Indeed, the term ‘orthogonal task’ is commonly used in some realms of cognitive neuroscience, especially in neuroimaging, to refer to a task not correlated with or independent from another condition of interest (Courtney 2012). In our case whether color targets A (more red) or B (more purple) were shown was not correlated with the orientation (45° or 135°) or visual field (LVF or RVF) of the concurrently displayed grating. We agree with the reviewer that this terminology may not be clear to a wider audience so we spell this relationship out more explicitly in the text.

“Importantly, the color task was orthogonal to the grating stimuli with the color targets displayed on a given trial being independent from the grating stimulus shown.”

3) The authors should provide more details how many leads of the multiwire array were used to test for prediction error responses, or whether they were combined into one multiunit record.

The results were from a single channel of the multi-wire array (channel 20). Only 8 of the 34 electrodes had enough SNR to record evoked responses from multi-unit

activity (see figure below). All of these electrodes exhibited a similar multi-unit activity profile in response to reward probability with varying levels of SNR. Therefore, we chose the channel with the best SNR to show the relationship between MUA and reward probability.

Adapted Figure Legend

“Peri-stimulus time histogram of normalized MUA recorded from a representative electrode from the chronic VTA array (see Methods).”

Adapted Methods

“From all the electrodes that had sufficient SNR to record MUA (8 of 34), we present data from a representative electrode with higher SNR although responses to reward predicting stimuli were consistent across electrodes.”

Point to Point Figure. Peri-stimulus time histogram of normalized MUA recorded from 8 electrodes with sufficient SNR from the chronic VTA array.

4) The authors should provide some reasoning why VTA-EM started 300 ms after visual stimulation?

The response to reviewer 1 question 7 is reproduced below:

Based on our results described in Arsenault et al. (2014), we have a strong hypothesis that associations between VTA-EM and a visual stimulus behave like an unconditioned (US) and conditioned stimulus (CS) in Pavlovian conditioning paradigms. Therefore, we chose forward conditioning [US (VTA-EM) starting after the start of the CS (visual stimulus)] over backward conditioning [US (VTA-EM) starting before the start of the CS (visual stimulus)] because:

1) backward conditioning can inhibit association learning (Siegel & Domjan 1974).

2) backward conditioning, with VTA-EM as the US, was shown to reduce cortical representations of CS while forward conditioning was shown to increase the cortical representation of the CS (Bao et al 2003).

In addition, we also chose delay [temporal overlapping of CS (visual stimulus) and US (VTA-EM)] over trace [a temporal delay between CS (visual stimulus) and US (VTA-EM)] forward conditioning. Delay conditioning was chosen because delay conditioning creates stronger associations than trace conditioning (Kryukov 2012). Therefore, we used a delay of VTA-EM (US) 300 ms after visual stimulus (CS) onset as a confluence of earlier decisions to **1)** use delay conditioning, **2)** 200 ms of VTA-EM and **3)** 500 ms visual stimulus.

As the reviewer suggested, we do expect that the timing of VTA-EM relative to the visual stimulus would affect the physiological and behavioral changes we monitored. The existing literature indicates a continuum moving from **A)** backward conditioning (CS before US) == reversed or inhibited learning/plasticity to **B)** delay forward conditioning (CS overlaps with US) == high learning/plasticity to **C)** trace forward conditioning (CS after US) == lower learning/plasticity). Based on this it would be highly interesting to examine how timing of the visual stimulus relative to VTA-EM affects learning/plasticity in our task-irrelevant learning paradigm. Despite this, the experiments undertaken in this study were extremely time-consuming. An average round of experiment 1 & 2 lasted for

16.5 x 3-4 hr. daily sessions or 22.5 days if weekends are included. Therefore, due to the time-consuming nature of these experiments we could not allocate the time needed for the admittedly interesting effects of stimulus-VTA-EM timing. Instead we focused on delay conditioning because we felt we would have the best chance to determine whether VTA-EM could generate task-irrelevant visual perceptual learning. To better address the rationale for timing the text below has been added to the results section.

“Because evidence suggests that delay conditioning creates relatively stronger associations between unconditioned and conditioned stimuli (Bao et al 2003, Kryukov 2012, Werden & Ross 1972), the 200 ms train of VTA-EM began 300 ms after the onset of the grating stimulus (Fig. 2b). Therefore, the conditioned stimulus (i.e. grating stimulus) preceded and overlapped with the unconditioned stimulus (i.e. VTA-EM).”

5) Why 100 Hz was used and why this may be effective, in particular if one considers the conduction velocity of dopaminergic fibers?

Our use of an 100 Hz for a stimulation frequency was based on the work of (Bao et al 2001) in which they examined the effect VTA-EM on frequency representations in the auditory cortex of rodents. This allowed us to examine whether the monkey visual system exhibited plasticity after cue-VTA-EM pairing like Bao et al., 2001 found in the auditory system of rodents. Looking at phasic responses of dopamine neurons in monkeys VTA-EM we find firing rates in the range of (10 Hz – 45 Hz) neurons but an example neurons with responses ~100 Hz have also been found [see Figure 3A from (Bayer & Glimcher 2005)]. In reference to the reviewers comment, dopamine neurons have been found to have a low conduction velocity which is in line with their low mean firing rate (Schultz & Romo 1987). Based on this mixed evidence we feel that 100 Hz is on the upper end of biologically plausible firing rates for the monkey VTA-EM. Despite this, it is entirely possible that 100 Hz is not the ideal frequency to stimulate the VTA in order to elicit stimulation induced learning. Because of this, we believe that stimulation

frequency is likely a crucial parameter in the parameter space that determines the strength of VTA stimulation induced plasticity. Nonetheless, a detailed examination of this parameter space is beyond the scope of this paper which instead was meant to generally examine whether VTA-EM could induce plasticity and learning. In addition, the amount of data needed to parametrically examine the effect of frequency on visual perceptual learning is far greater than the substantially large dataset already acquired for this study.

6) Was the stimulation unipolar or bipolar?

Stimulating electrodes were unipolar with stimulation being performed between stimulating electrodes in the VTA and a low impedance ground electrode. We have amended the methods to reflect this important attribute of the experimental preparation.

“Unipolar electrical microstimulation was performed using the stimulating electrodes in the VTA and a low impedance ground wire implanted below the skull.”

Point to Point References

- Amano K, Shibata K, Kawato M, Sasaki Y, Watanabe T. 2016. Learning to Associate Orientation with Color in Early Visual Areas by Associative Decoded fMRI Neurofeedback. *Curr Biol* 26: 1861-6
- Arsenault JT, Rima S, Stemmann H, Vanduffel W. 2014. Role of the primate ventral tegmental area in reinforcement and motivation. *Curr Biol* 24: 1347-53
- Bao S, Chan VT, Merzenich MM. 2001. Cortical remodelling induced by activity of ventral tegmental dopamine neurons. *Nature* 412: 79-83
- Bao S, Chan VT, Zhang LI, Merzenich MM. 2003. Suppression of cortical representation through backward conditioning. *Proc Natl Acad Sci U S A* 100: 1405-8
- Bates D, Maechler M, Bolker B, Walker S. 2015. Fitting Linear Mixed-Effects Models Using lme4. *Journal of Statistical Software* 67: 1-48
- Bayer HM, Glimcher PW. 2005. Midbrain dopamine neurons encode a quantitative reward prediction error signal. *Neuron* 47: 129-41
- Berger B, Gaspar P, Verney C. 1991. Dopaminergic innervation of the cerebral cortex: unexpected differences between rodents and primates. *Trends Neurosci* 14: 21-7
- Berrios J, Stamatakis AM, Kantak PA, McElligott ZA, Judson MC, et al. 2016. Loss of UBE3A from TH-expressing neurons suppresses GABA co-release and enhances VTA-NAc optical self-stimulation. *Nat Commun* 7: 10702
- Bourgeois A, Chelazzi L, Vuilleumier P. 2016. How motivation and reward learning modulate selective attention. *Prog Brain Res* 229: 325-42
- Bromberg-Martin ES, Matsumoto M, Hikosaka O. 2010. Dopamine in motivational control: rewarding, aversive, and alerting. *Neuron* 68: 815-34
- Censor N, Sagi D, Cohen LG. 2012. Common mechanisms of human perceptual and motor learning. *Nat Rev Neurosci* 13: 658-64
- Chubykin AA, Roach EB, Bear MF, Shuler MG. 2013. A cholinergic mechanism for reward timing within primary visual cortex. *Neuron* 77: 723-35
- Courtney SM. 2012. Development of orthogonal task designs in fMRI studies of higher cognition: the NIMH experience. *Neuroimage* 62: 1185-9
- Ekstrom LB, Roelfsema PR, Arsenault JT, Bonmassar G, Vanduffel W. 2008. Bottom-up dependent gating of frontal signals in early visual cortex. *Science* 321: 414-7
- Froemke RC, Carcea I, Barker AJ, Yuan K, Seybold BA, et al. 2013. Long-term modification of cortical synapses improves sensory perception. *Nat Neurosci* 16: 79-88
- Gattass R, Galkin TW, Desimone R, Ungerleider LG. 2014. Subcortical connections of area V4 in the macaque. *J Comp Neurol* 522: 1941-65
- Gerits A, Farivar R, Rosen BR, Wald LL, Boyden ES, Vanduffel W. 2012. Optogenetically induced behavioral and functional network changes in primates. *Curr Biol* 22: 1722-6
- Gerits A, Vanduffel W. 2013. Optogenetics in primates: a shining future? *Trends Genet* 29: 403-11
- Histed MH, Bonin V, Reid RC. 2009. Direct activation of sparse, distributed populations of cortical neurons by electrical microstimulation. *Neuron* 63: 508-22
- Histed MH, Ni AM, Maunsell JH. 2013. Insights into cortical mechanisms of behavior from microstimulation experiments. *Prog Neurobiol* 103: 115-30
- Huerta MF, Krubitzer LA, Kaas JH. 1987. Frontal eye field as defined by intracortical microstimulation in squirrel monkeys, owl monkeys, and macaque monkeys. II. Cortical connections. *J Comp Neurol* 265: 332-61
- Janssens T, Zhu Q, Popivanov ID, Vanduffel W. 2014. Probabilistic and single-subject retinotopic maps reveal the topographic organization of face patches in the macaque cortex. *J Neurosci* 34: 10156-67
- Kilgard MP, Merzenich MM. 1998. Cortical map reorganization enabled by nucleus basalis activity. *Science* 279: 1714-8
- Kosofsky BE, Molliver ME, Morrison JH, Foote SL. 1984. The serotonin and norepinephrine innervation of primary visual cortex in the cynomolgus monkey (*Macaca fascicularis*). *J Comp Neurol* 230: 168-78
- Kryukov VI. 2012. Towards a unified model of pavlovian conditioning: short review of trace conditioning models. *Cogn Neurodyn* 6: 377-98

- Liu CH, Coleman JE, Davoudi H, Zhang K, Hussain Shuler MG. 2015. Selective activation of a putative reinforcement signal conditions cued interval timing in primary visual cortex. *Curr Biol* 25: 1551-61
- Morales M, Margolis EB. 2017. Ventral tegmental area: cellular heterogeneity, connectivity and behaviour. *Nat Rev Neurosci* 18: 73-85
- Noudoost B, Moore T. 2011. Control of visual cortical signals by prefrontal dopamine. *Nature* 474: 372-5
- Pinto L, Goard MJ, Estandian D, Xu M, Kwan AC, et al. 2013. Fast modulation of visual perception by basal forebrain cholinergic neurons. *Nat Neurosci* 16: 1857-63
- Premereur E, Van Dromme IC, Romero MC, Vanduffel W, Janssen P. 2015. Effective connectivity of depth-structure-selective patches in the lateral bank of the macaque intraparietal sulcus. *PLoS Biol* 13: e1002072
- Prusky GT, Douglas RM. 2004. Characterization of mouse cortical spatial vision. *Vision Res* 44: 3411-8
- Reed A, Riley J, Carraway R, Carrasco A, Perez C, et al. 2011. Cortical map plasticity improves learning but is not necessary for improved performance. *Neuron* 70: 121-31
- Sander CY, Arsenault JT, Rosen BR, J.B. M, Vanduffel W. 2018. Functional signaling contributions of D1 and D2 dopamine receptors due to VTA stimulation in non-human primates. *International Society for Magnetic Resonance in Medicine*
- Schall JD, Morel A, King DJ, Bullier J. 1995. Topography of visual cortex connections with frontal eye field in macaque: convergence and segregation of processing streams. *J Neurosci* 15: 4464-87
- Schluter EW, Mitz AR, Cheer JF, Averbek BB. 2014. Real-time dopamine measurement in awake monkeys. *PLoS One* 9: e98692
- Schultz W, Romo R. 1987. Responses of nigrostriatal dopamine neurons to high-intensity somatosensory stimulation in the anesthetized monkey. *J Neurophysiol* 57: 201-17
- Seitz AR, Kim D, Watanabe T. 2009. Rewards evoke learning of unconsciously processed visual stimuli in adult humans. *Neuron* 61: 700-7
- Shibata K, Watanabe T, Sasaki Y, Kawato M. 2011. Perceptual learning incepted by decoded fMRI neurofeedback without stimulus presentation. *Science* 334: 1413-5
- Shuler MG, Bear MF. 2006. Reward timing in the primary visual cortex. *Science* 311: 1606-9
- Siegel S, Domjan M. 1974. Inhibitory Effect of Backward Conditioning as a Function of Number of Backward Pairings. *B Psychonomic Soc* 4: 122-24
- Stauffer WR, Lak A, Yang A, Borel M, Paulsen O, et al. 2016. Dopamine Neuron-Specific Optogenetic Stimulation in Rhesus Macaques. *Cell* 166: 1564-71 e6
- Tolias AS, Sultan F, Augath M, Oeltermann A, Tehovnik EJ, et al. 2005. Mapping cortical activity elicited with electrical microstimulation using FMRI in the macaque. *Neuron* 48: 901-11
- van Kerkoerle T, Marik SA, Meyer Zum Alten Borgloh S, Gilbert CD. 2018. Axonal plasticity associated with perceptual learning in adult macaque primary visual cortex. *Proc Natl Acad Sci U S A* 115: 10464-69
- Werden D, Ross LE. 1972. A comparison of the trace and delay classical conditioning performance of normal children. *J Exp Child Psychol* 14: 126-32
- Yamahachi H, Marik SA, McManus JN, Denk W, Gilbert CD. 2009. Rapid axonal sprouting and pruning accompany functional reorganization in primary visual cortex. *Neuron* 64: 719-29
- Zhang S, Qi J, Li X, Wang HL, Britt JP, et al. 2015. Dopaminergic and glutamatergic microdomains in a subset of rodent mesoaccumbens axons. *Nat Neurosci* 18: 386-92

REVIEWERS' COMMENTS:

Reviewer #1 (Remarks to the Author):

The authors have done a great job addressing my concerns. No further comments.

Reviewer #2 (Remarks to the Author):

Arsenault and Vanduffel have substantially improved the data analyses and the description of experimental details, have significantly expanded the discussion and considered relevant literature much more intensively. The authors point out that this study provides the first causal evidence that the activation of primate neuromodulatory centers can drive perceptual learning. Although this research is highly relevant to a wider audience and the results are sound, it does not fully become clear in which respect this study provides a major advance over the existing knowledge on perceptual learning, in particular that obtained from rodents. It is not sufficient to make the unspecific statement that compared to primates rodents have reduced visual acuity and weaker cortical dopaminergic innervation.

REVIEWERS' COMMENTS:

Reviewer #1 (Remarks to the Author):

The authors have done a great job addressing my concerns. No further comments.

Reviewer #2 (Remarks to the Author):

Arsenault and Vanduffel have substantially improved the data analyses and the description of experimental details, have significantly expanded the discussion and considered relevant literature much more intensively. The authors point out that this study provides the first causal evidence that the activation of primate neuromodulatory centers can drive perceptual learning. Although this research is highly relevant to a wider audience and the results are sound, it does not fully become clear in which respect this study provides a major advance over the existing knowledge on perceptual learning, in particular that obtained from rodents. It is not sufficient to make the unspecific statement that compared to primates rodents have reduced visual acuity and weaker cortical dopaminergic innervation.

The visual system of the macaque is substantially more similar to the human visual system than the rodent. This is not surprising given that both human and monkeys are genetically more similar, and both species rely more heavily on vision. In addition, the cortical distribution of dopamine has diverged significantly between primates and rodents, which is clearly evident in visual areas (see table below reproduced from Berger et al. 1991). Therefore, we feel our research provides major advances on the understanding of dopaminergic centers role in primate visual plasticity and visual perceptual learning. Moreover, the work on task-irrelevant learning in humans demonstrates that while subjects actively perform a visual task, visual perceptual learning of irrelevant features occurs when these features are correlated with reinforcement. To explore this phenomenon in primates, we had to train monkeys to fixate centrally while performing a difficult peripheral color discrimination task. In addition, we needed to train the animals to perform a separate motion discrimination task and to switch back and forth between these difficult tasks. Moreover, monkeys performed these tasks at a level comparable and most often better than humans. Visual tasks of this difficulty are straightforward in humans and frankly not possible in rodents. Therefore, we feel our work bridges the gap between rodent and human studies, providing causal evidence for the role of the VTA in this type of visual perceptual learning. Accordingly, we have added two sentences to the discussion to indicate this.

“In addition, during this study monkeys alternated between difficult visual tasks involving sustained fixation, peripheral attention and hand responses in a manner that was

comparable to human studies of task-irrelevant VPL. Therefore, our work bridges the gap between rodent and human investigations of VPL.”

TABLE I. Comparative regional and laminar distribution of DA afferents to the cerebral cortex

	Rodents						Primates					
	I	II	III	IV	V	VI	I	II	III	IV	V	VI
Prefrontal	•	•	•		•	•	•	•	•	•	•	•
Anterior cingulate	•	•	•		•	•	•	•	•	•	•	•
Insula agranular	•	•	•		•	•	•	•	•		•	•
Entorhinal	•	•	•		•	•	•	•	•		•	•
Motor						•	•	•	•		•	•
Premotor	•	•	•		•	•	•	•	•		•	•
Supplementary motor	•	•	•		•	•	•	•	•		•	•
Parietal						•	•	•	•	•	•	•
Temporal						•	•	•	•	•	•	•
Visual	•	•	•			•	•	•	•		•	•

The black dots of different sizes give a rough representation of the relative laminar density of DA afferents for each species, but not between the two species.

Table 1 reproduced from Berger et al. 1991.